

# Sea Ice Albedo Bounded Data Assimilation and Its Impact on Modeling: A Regional Approach

Joseph F. Rotondo[1], Molly M. Wieringa[1,2], Cecilia M. Bitz[1], Robin Clancy[3], and Steven M. Cavallo[3]

[1]Department of Atmospheric and Climate Science, University of Washington, Seattle, Washington
[2]Advanced Study Program, NSF National Center for Atmospheric Research, Boulder, Colorado
[3]School of Meteorology, University of Oklahoma, Norman, Oklahoma

**Correspondence:** Joseph F. Rotondo (jrotondo@uw.edu)

**Abstract.** We conducted a perfect model experiment using `Icepack`, a one-dimensional single-column sea ice model, to assess the potential of data assimilation (DA) to improve predictions of the mean sea ice state through the incorporation of sea ice albedo (SIAL) observations. One ensemble member is designated as the TRUTH, and synthetic observations drawn from it are assimilated into the remaining ensemble members. DA is carried out using the Data Assimilation Research Testbed (DART) with a bounded Quantile Conserving Ensemble Filtering Framework (QCEFF), which accounts for the bounded nature of sea ice variables. `Icepack` ensembles were spun-up for four Arctic locations based on small perturbations to atmospheric forcing. Results show that assimilating SIAL yields comparable or superior performance to more commonly assimilated observables such as sea ice concentration (SIC) and thickness (SIT) in three-quarters of the Arctic regions studied, and across all regions when observational uncertainty in SIAL is reduced below estimates from the current literature. These findings underscore the value of leveraging existing SIAL observations and expanding their temporal and spatial coverage in the Arctic. Furthermore, the study highlights the critical need to better constrain the observational uncertainty of SIAL. Enhanced observational networks would provide the necessary validation data, enabling more accurate uncertainty characterization and improved sea ice forecasts in a rapidly evolving polar climate.

## 1 Introduction

Arctic amplification (AA) refers to the phenomenon in which the Arctic experiences accelerated warming compared to the global average. This enhanced warming has been consistently observed over recent decades (Rantanen et al., 2022). The accelerated warming is partially attributed to the ice-albedo feedback mechanism, in which the melting of snow and sea ice exposes darker underlying surfaces, enhancing the absorption of short-wave radiation and further amplifying ice loss (Perovich and Polashenski, 2012; Serreze and Barry, 2011). Despite the critical role of this feedback in the dynamics of Arctic climate, global climate models (GCMs) exhibit persistent limitations in accurately simulating Arctic sea ice. These deficiencies contribute to significant uncertainties in the projection of key sea ice variables, including sea ice concentration (SIC), sea ice thickness (SIT), and surface albedo (Donohoe et al., 2020; Pithan and Mauritsen, 2014).

Sea ice albedo (SIAL), defined as the fraction of incoming solar radiation reflected back into the atmosphere over the ice-covered portion of a grid cell, is a critical component of the polar climate system's energy budget. Unlike broadband surface



albedo, SIAL in climate models is calculated independently of sea ice concentration (SIC), in that it excludes contributions
from open water and characterizes only the reflectivity of sea ice itself. However, in practice, SIAL and SIC often co-vary in
response to shared surface processes such as melt ponding, snow loss, and sea ice retreat. Bare sea ice exhibits a relatively
high albedo, typically reflecting 50–70% of incident short-wave radiation, in stark contrast to the open ocean, which only
reflects about 10%, absorbing most of the incoming short-wave radiation due to its much lower albedo. Snow-covered sea ice

can reflect 90% of incoming short-wave radiation, whereas melt ponds, which form on the ice surface during the melt season,
exhibit highly variable albedo values. These values depend on factors such as pond depth and water clarity, often resulting
in significantly lower albedos relative to bare ice (Grenfell and Perovich, 2004; Perovich, 1996). The spatial and temporal
variability of SIAL–driven by changes in surface conditions on the sea ice, such as snow cover, bare ice, and melt ponding–
modulates the amount of solar radiation reflected within ice–covered regions. SIAL is typically derived from satellite remote

sensing, which enables spatially extensive, long-term observations, but it is also measured directly via in-situ instruments
deployed on the ice, offering higher-resolution insights into local surface conditions (Karlsson et al., 2023; Calmer et al., 2023).
While the primary ice–albedo feedback arises from the strong contrast between sea ice and open ocean albedos, variability in
SIAL governs the intensity and spatial distribution of this feedback over ice-covered areas. Accordingly, SIAL serves as an
important regulator of the Arctic shortwave radiative balance, particularly during the summer melt season (Seong et al., 2022;

Arndt and Nicolaus, 2014).

The reduction in surface albedo from melting sea ice is complicated by secondary processes, such as increased cloud cover
and rising ocean temperatures, which compound disruptions to the energy balance and either intensify or dampen the cycle of
Arctic warming (Sledd and L'Ecuyer, 2021; Taylor et al., 2015). As a result, capturing these intricate dynamics within coupled
sea ice simulations is vital for projecting both seasonal and long-term sea ice trends and for understanding their broader

implications on Arctic and global climate systems. The authors leave this as an open area of study for future research based on
improved representation of Arctic sea ice.

The primary aim of this study is to improve the accuracy of sea ice simulations by integrating idealized albedo observa-
tions within an ensemble data assimilation (DA) framework using synthetic (perfect model) experiments. Assimilating albedo
observations offers a pathway to refine the representation of modeled energy exchange processes, reducing uncertainties in

simulations of Arctic sea ice. While this study serves as a proof-of-concept demonstration under idealized conditions, such
an approach has the potential to inform future improvements in model parameterizations related to melt thermodynamics, ice
growth, and ice thickness distributions (Hunke et al., 2010; Lindsay, 2001; Barry, 1996).

The inclusion of SIAL in the DA framework is expected to improve the ability of the model to capture interannual variability
and very rapid sea ice loss events (VRILEs), which are essential to understand the rapid sea ice transformations that occur in

the Arctic (Cavallo et al., 2025; Pistone et al., 2014; Schröder et al., 2014; Perovich et al., 2008). By addressing deficiencies
in SIAL representation through idealized DA experiments, this study aims to lay the groundwork for future improvements in
subseasonal-to-decadal sea ice forecasting skill and projections of Arctic sea ice trends. The integration of SIAL observations
enhances the accuracy of model state estimates–helping to better constrain simulations without altering the model's physical





formulation. While this study does not directly assess ecosystem or climate impacts, improving the representation of albedo
may ultimately support broader efforts to reduce uncertainty in Arctic climate variability and its global feedbacks.

## 2   Methods

To quantify the impact of SIAL DA, we conducted a series of perfect-model experiments using the `Icepack` sea ice model
(v1.4.0; (Hunke et al., 2023)), the thermodynamic module of the widely used Community Ice CodE (CICE). `Icepack` is
a one-dimensional, column-based model that simulates vertical thermodynamic processes in sea ice but does not include
horizontal ice dynamics such as advection. For this study, `Icepack` was spun up from 2000 to 2010 to establish realistic local
ice conditions in four key Arctic regions: the Barents Sea (75° N, 40° E), the Central Arctic (88° N, 0° E), Coastal Canada
(81° N, 2° W), and the Siberian–Chukchi Sea (76° N, 174° E). This 11-year spin-up period allows the model to equilibrate
to prescribed atmospheric and oceanic forcing, eliminating artifacts from idealized initial conditions and ensuring that all
subsequent simulations begin from physically consistent and realistic ice states. The early 2000s were selected as a critical
transition period in Arctic sea ice history, during which the region began shifting from a predominantly multi-year ice cover
to one dominated by first-year ice (Sumata et al., 2023). Each `Icepack` simulation was forced with prescribed atmospheric
conditions from the Japanese 55-year Reanalysis for driving ocean–sea-ice models (JRA-55-do) and coupled to a slab ocean
following Tsujino et al. (2018). Initial conditions for the slab ocean were derived from the ocean component output of a fully
coupled historical simulation using the Community Earth System Model (CESM2). For each region, we generated an ensemble
of 30 `Icepack` simulations to capture internal variability and provide a basis for subsequent DA experiments.

   `Icepack` parameterizes the ice thickness distribution (ITD) by discretizing the continuous range of sea ice thicknesses
into $n$ distinct thickness categories (with $n = 5$ in our integrations). The model state variables–$aice_n$ (ice area), $vice_n$ (ice
volume), and $vsno_n$ (snow volume)–are each defined per thickness category. This categorization allows the model to capture
the heterogeneous and nonlinear behavior of sea ice processes across the thickness spectrum, enabling physically consistent
thermodynamics and parameterized mechanical interactions (e.g., ridging and rafting) for thin versus thick ice.

   We selected geographic locations (Figure 1) to assess how SIAL assimilation influences the modeled sea ice behavior in
different Arctic regions, with a particular focus on how local atmospheric forcings shape ice evolution in areas with varying
sea ice climatologies (Tschudi et al., 2020; Serreze and Barry, 2011). We overlay mean annual SIC (2000-2015) from the
National Snow and Ice Data Center (NSIDC) Climate Data Record (CDR) for a greater understanding of typical ice conditions
at these four locations (Meier et al., 2024).

   The spun-up ensembles were then integrated forward for a 5-year period (2011–2015) without any DA. The 5-year period
following spin-up was selected to balance two goals: ensuring a long enough time window to evaluate the cumulative impact
of DA on sea ice state evolution, while remaining short enough to minimize the influence of structural model drift and changes
in climate forcing not represented by the perfect-model framework. This period also aligns with the availability of well-
characterized reanalysis inputs and avoids strong interannual anomalies that could dominate the signal. Additionally, using a
relatively short post-spin-up period helps isolate the effects of initial condition uncertainty and DA rather than external forcing





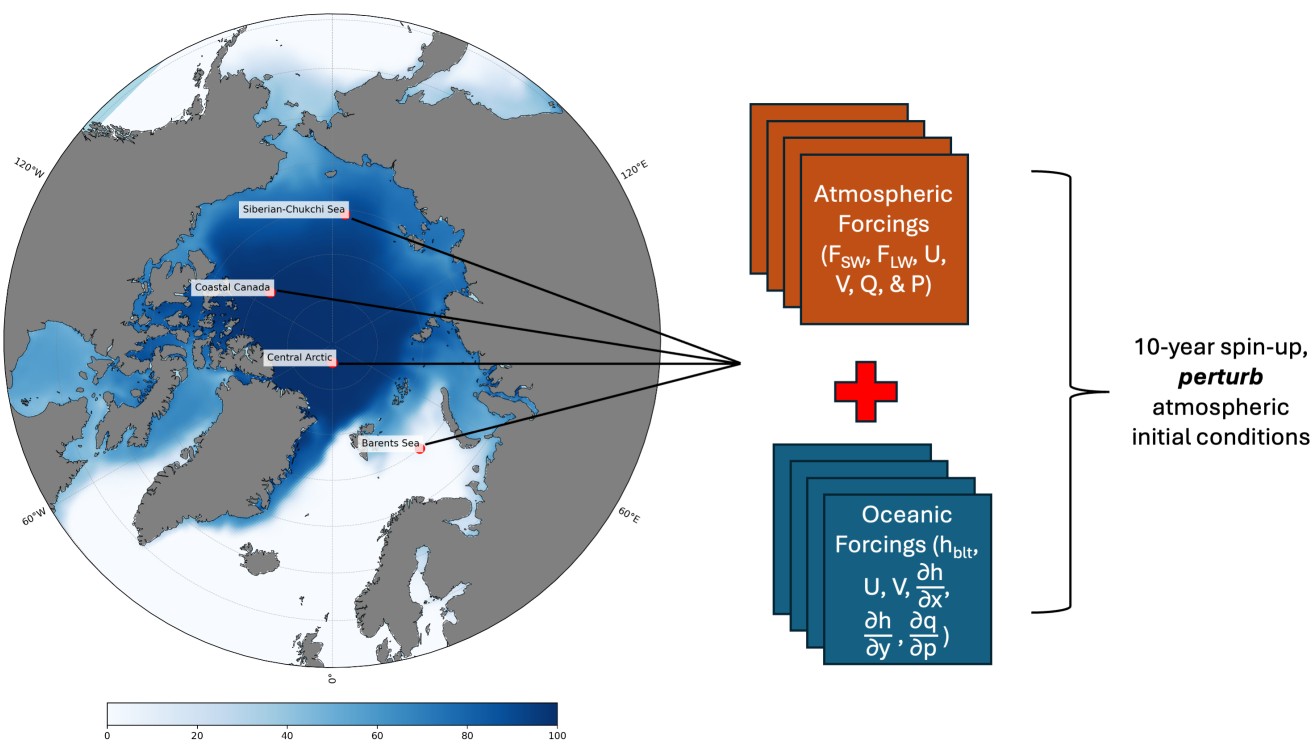

**Figure 1.** Geographic locations of the atmospheric and oceanic forcing selected for analysis. Atmospheric forcings used to spin up the sea ice ensemble include downwelling shortwave ($F_{SW}$) and longwave ($F_{LW}$) radiation (W m$^{-2}$), 10-meter zonal ($U$) and meridional ($V$) wind speeds (m s$^{-1}$), specific humidity ($Q$, kg kg$^{-1}$), and precipitation ($P$, kg m$^{-2}$ s$^{-1}$). Oceanic (slab) forcings include temperature ($T$, K), salinity ($S$, PSU), mixed layer depth ($h_{blt}$, m), surface zonal and meridional currents ($U$, $V$, m s$^{-1}$), ocean surface tilt ($\partial h/\partial x$, $\partial h/\partial y$; unitless), and vertical convergence of heat transport ($\partial q/\partial p$, W m$^{-3}$).

trends. This free run serves as a control case against which assimilation experiments can be evaluated. For each assimilation experiment–defined as the set of simulations conducted for one of the four Arctic locations–a randomly selected ensemble member was designated as the reference TRUTH state, from which synthetic observations were derived for assimilation.

To account for sensitivity to the choice of TRUTH, we repeated the assimilation experiments using ten different ensemble members as TRUTH states (ensemble members 3, 5, 10, 12, 14, 16, 19, 21, 25, and 28). Synthetic observations of SIAL, SIC, and SIT were generated from each of these TRUTH realizations for assimilation into the remaining ensemble members. Using synthetic observations is advantageous as they share the same spatial and temporal scales as the model. Moreover, in a one-dimensional framework, these observations can be assimilated and tested for significance with substantially lower

computational cost compared to assimilating real observations that are not co-located with the model column.

Figure 2 presents the ensemble mean ($\mu$) SIC in the free run at each selected location for 2011–2015, along with the ensemble spread indicated by ensemble standard deviation ($\pm 2\sigma$). The Barents Sea location exhibits the highest variability in SIC across





the study period and does not reach full ice coverage in the ensemble mean. The Siberian–Chukchi Sea also displays a highly
seasonal SIC cycle, achieving near-complete coverage during winter but retreating substantially in summer. Both regions lie
within the marginal ice zone (MIZ), where sea ice frequently transitions between open water and partial coverage. To represent
these dynamic conditions, we configure these sites in `Icepack` using the *fluxing open water* boundary condition. In standalone
`Icepack` simulations, this functionality is required to approximate the exchange of energy and mass between sea ice and the
surrounding ocean, particularly in regions with seasonally variable ice cover. Without a dynamic coupling to an ocean model,
`Icepack` must either assume uniformly ice-covered conditions (*fluxing uniform ice*) or allow for partially open water that
can receive and exchange fluxes at the ice edge. The *fluxing open water* option provides a more realistic treatment of MIZ
dynamics in these seasonally ice-covered regions.

The Coastal Canada region exhibits a pronounced seasonal signal, but remains partially ice covered even during the summer
minimum. Because `Icepack` does not include coastlines or ice advection, it cannot capture the coastal ice buildup that
is well-represented in observations and full three-dimensional sea ice models. As a result, SIC values in this region are likely
underestimated. In contrast, the Central Arctic is dominated by perennial, multi-year ice and shows minimal seasonal variability
in SIC, making it a stable reference point for comparison. These two sites are therefore configured to represent *fluxing uniform
ice*—locations where sea ice dynamics are primarily governed by internal redistribution and deformation within a contiguous
ice pack, rather than interactions with open water or ice edge processes.

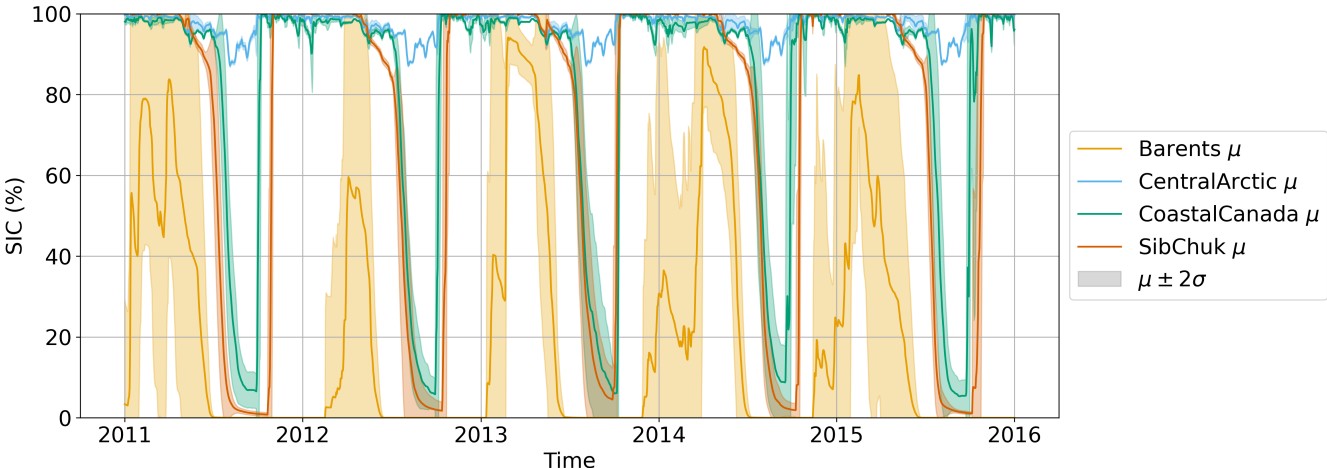

**Figure 2.** Mean ($\mu$) and standard deviation ($\pm 2\sigma$) of SIC across ensemble members and region of the `Icepack` free-run for 2011-2015.

## 2.1  Data Assimilation Setup

One of the most challenging aspects of sea ice DA is the inherent boundedness of the modeled variables. Quantities such
as SIAL (bounded between 0 and 1), SIC (modeled as a fraction from 0 to 1, reported here as 0–100%), and SIT ($\geq 0$ m)
pose limitations on the potential ensemble spread, especially during periods when these variables approach their physical





bounds—typically in winter (near-maximum values for SIC) and summer (near-minimum values for SIC and SIT). During transition seasons, SIAL and SIC can also exhibit rapid nonlinear changes due to melt pond formation, refreezing, snowfall,

and other surface processes. In such cases, the model ensemble may lack sufficient variability to adequately represent fast seasonal transitions, making it difficult for traditional DA methods, many of which rely on unbounded Gaussian assumptions, to optimally update the model state. These limitations necessitate alternative assimilation frameworks that explicitly account for physical bounds and distribution asymmetries.

To address these challenges, we employed the Quantile Conserving Ensemble Filtering Framework (QCEFF; (Anderson,

2022, 2023; Anderson et al., 2024)), implemented within the Data Assimilation Research Testbed (DART; Anderson et al., 2009). DART is a community DA system developed by the Data Assimilation Research Section (DAReS) of the National Center for Atmospheric Research (NCAR) with the support of the National Science Foundation (NSF). Specifically, we used the bounded normal variant of the Rank Histogram Filter (RHF), which combines the statistical rigor of Gaussian-based assimilation with the physical realism of bounded state variables. The filter works by fitting a truncated normal distribution to

the ensemble, conserving rank-based quantiles during the update step while ensuring that the resulting values remain within physically plausible limits. This approach has been shown to improve the performance of DA systems when dealing with constrained geophysical variables, as it prevents unphysical ensemble updates (e.g., SIAL > 1 or negative SIT; (Wieringa et al., 2024; Anderson et al., 2024; Riedel et al., 2025)).

SIAL assimilation, in particular, benefits substantially from this bounded framework. Unlike SIC and SIT, which frequently

approach their lower or upper physical limits, SIAL values typically remain within a central range–rarely falling below 0.1 or exceeding 0.9–even during extreme seasonal transitions. This characteristic makes SIAL an ideal candidate for assimilation within a truncated Gaussian filter, as the ensemble spread is more likely to encompass the true state without frequently encountering hard boundaries. As a result, the bounded RHF can fully leverage its quantile-conserving properties without being regularly constrained by the extremes of the distribution.

While this bounded formulation has recently been applied to SIC and SIT (e.g., Riedel et al., 2025; Wieringa et al., 2024), its implementation for SIAL assimilation is novel. Assimilating in `Icepack` with the QCEFF method allows for a more consistent and physically grounded incorporation of SIAL and other bounded observations, underscoring the potential of bounded DA algorithms to advance the prediction capabilities of sea ice systems.

## 2.2 Observations

Synthetic observations assimilated by DART are generated by applying random noise drawn from a non-Gaussian bounded distribution with a standard deviation of $2\sigma$, centered on the TRUTH value. A sensitivity test examining the influence of random observational noise realizations on assimilation performance is included in the Supplement (Figure S1). This process accounts for uncertainties that would affect real-world observations of the selected variables (Anderson et al., 2024). The synthetic observations are calculated as aggregates of modeled quantities categorized by thickness in `Icepack` (e.g., $aice_n$, $vice_n$) over

these thickness categories. Most in-situ observations, by contrast, are point measurements of single variables such as SIT or SIC, which are typically treated as diagnostic quantities in sea ice models. In one experiment, we depart from this aggregate



framework and instead assimilate synthetic SIAL observations separately within each thickness category–that is, we provide the DA system with SIAL values corresponding to each modeled category, $SIAL_n$. This allows us to investigate the role of modeled SIAL across the thickness categories in assimilation performance, particularly in the Siberian–Chukchi Sea region, where the standard approach underperforms.

The synthetic observation types used in our DA experiments include SIC, SIT, and four SIAL components derived from `Icepack`'s narrow-band albedo scheme: direct visible ($\alpha_{\mathrm{DirVis}}$), direct infrared ($\alpha_{\mathrm{DirIR}}$), indirect visible ($\alpha_{\mathrm{IndVis}}$), and indirect infrared ($\alpha_{\mathrm{IndIR}}$). These observations are calculated as aggregates over the model's thickness categories based on quantities such as $aice_n$ and $vice_n$. In our configuration, which uses `Icepack`'s 3-band Delta-Eddington radiative transfer scheme (`shortwave = 'dEdd'`), direct and indirect shortwave radiation are treated equivalently, and so albedo observations are grouped into two spectral bands—visible ($\alpha_{\mathrm{Vis}}$) and infrared ($\alpha_{\mathrm{IR}}$)—for analysis. While `Icepack` supports a more spectrally resolved 5-band scheme, it was not used in this study. The simplification is consistent with both the 3-band scheme's structure and the high correlation observed between direct and indirect albedos in version 1.4.0 (CICE Consortium, 2025).

The prescribed observational uncertainty distributions for each synthetic observation are summarized in Table 1 and visualized in Appendix Figure A1. The synthetic observational uncertainties listed in Table 1 were specified based on values reported in previous literature. For the narrow-band aggregate SIAL observations ($\alpha_{\mathrm{DirVis}}$, $\alpha_{\mathrm{DirIR}}$, $\alpha_{\mathrm{IndVis}}$, and $\alpha_{\mathrm{IndIR}}$), three levels of observational uncertainty were adopted, informed by estimates from Riihelä et al. (2024) and Xiong et al. (2002). These three uncertainty levels—$\pm5\%$, $\pm14\%$, and $\pm25\%$—reflect the limited validation data available for satellite-derived SIAL observations and are intended to represent low, medium, and high uncertainty scenarios, respectively. The few existing in-situ validation studies suggest that actual satellite observational retrieval uncertainties are likely closer to the low-to-medium uncertainty range (Riihelä et al., 2010; Xiong et al., 2002).

The observational uncertainty for aggregate SIC is modeled as a negative parabola (Table 1), with the greatest uncertainty occurring within the MIZ, where SIC ranges between 15% and 85%. This formulation reflects current understanding that satellite retrievals of very low or very high SIC are generally more accurate than those within the MIZ, where spatial heterogeneity and measurement limitations introduce greater uncertainty (Wernecke et al., 2024; Han et al., 2021; Brucker et al., 2014). It is important to note that this uncertainty parameterization can also depend on surface conditions such as snow cover and the presence of polynyas, which are not explicitly included in our uncertainty calculations, but are instead solely dependent on SIC.

Similarly, the observational retrieval uncertainty for aggregate SIT is informed by satellite observation campaigns such as ICESat-2 and CryoSat-2. For simplicity, the SIT uncertainty is approximated as 10% of the observed value (Table 1), which likely underestimates the true observational uncertainty and should be interpreted as a lower bound, consistent with prior estimates (Zhang et al., 2023; Stonebridge et al., 2018). It should be noted that this relationship may not hold at very low SIT values, where sea ice freeboard becomes small or even negative, introducing greater observational challenges (Rösel et al., 2018). Additionally, this somewhat arbitrary choice of uncertainty is likely optimistic–uncertainties from older SIT observational datasets are considerably larger (e.g., Schweiger et al. 2011).





| Observation Type | Symbol | Forward Operator (Aggregate Equation) | Prescribed Observational Uncertainty ($2\sigma$) |
|---|---|---|---|
| Sea Ice Concentration | SIC | $\mathrm{SIC_{agg}} = \sum_{n=1}^{n_{\mathrm{cat}}} aice_n$ | $-\frac{1}{2}(\mathrm{SIC}^2 - \mathrm{SIC})$ |
| Sea Ice Thickness | SIT | $\mathrm{SIT_{agg}} = \dfrac{\sum_{n=1}^{n_{\mathrm{cat}}} \mathrm{vice}_n}{\mathrm{SIC_{agg}}}$ <br> valid only where $\mathrm{SIC_{agg}} > 0$ | $0.1 \times \mathrm{SIT}$ |
| Direct Visible SIAL | $\alpha_{\mathrm{DirVis}}$ | $\alpha_{\mathrm{DirVis,agg}} = \dfrac{\sum_{n=1}^{n_{\mathrm{cat}}} (\alpha_{\mathrm{DirVis},n} \cdot aice_n)}{\mathrm{SIC_{agg}}}$ | $0.05 \times \alpha$ (low) <br> $0.14 \times \alpha$ (medium) <br> $0.25 \times \alpha$ (high) |
| Direct Infrared SIAL | $\alpha_{\mathrm{DirIR}}$ | $\alpha_{\mathrm{DirIR,agg}} = \dfrac{\sum_{n=1}^{n_{\mathrm{cat}}} (\alpha_{\mathrm{DirIR},n} \cdot aice_n)}{\mathrm{SIC_{agg}}}$ | $0.05 \times \alpha$ (low) <br> $0.14 \times \alpha$ (medium) <br> $0.25 \times \alpha$ (high) |
| Indirect Visible SIAL | $\alpha_{\mathrm{IndVis}}$ | $\alpha_{\mathrm{IndVis,agg}} = \dfrac{\sum_{n=1}^{n_{\mathrm{cat}}} (\alpha_{\mathrm{IndVis},n} \cdot aice_n)}{\mathrm{SIC_{agg}}}$ | $0.05 \times \alpha$ (low) <br> $0.14 \times \alpha$ (medium) <br> $0.25 \times \alpha$ (high) |
| Indirect Infrared SIAL | $\alpha_{\mathrm{IndIR}}$ | $\alpha_{\mathrm{IndIR,agg}} = \dfrac{\sum_{n=1}^{n_{\mathrm{cat}}} (\alpha_{\mathrm{IndIR},n} \cdot aice_n)}{\mathrm{SIC_{agg}}}$ | $0.05 \times \alpha$ (low) <br> $0.14 \times \alpha$ (medium) <br> $0.25 \times \alpha$ (high) |

**Table 1.** Observation types, forward operators, and prescribed observational uncertainty assumptions. SIAL aggregates are weighted by sea ice area and normalized by SIC, making them representative of ice-covered portions of each grid cell (i.e., valid only where SIC > 0).

## 2.3 Assimilation Temporal Selection

The DA methodology involved assimilating daily synthetic observations from April 1 to October 15, 2011. The temporal range–from spring through early autumn–was selected to align with the period during which satellite-derived SIAL observations are available and solar radiation effectively reaches the high Arctic, enabling reliable SIAL retrievals. This also coincides with the season when SIAL plays a key radiative role; during winter, limited solar insolation renders SIAL variations largely inconsequential to the surface energy budget.

Daily observational SIAL products are available from the Polar Pathfinder (APP-x) dataset, which uses data from the Advanced Very High Resolution Radiometer (AVHRR) sensor (Tschudi et al., 2019). Consequently, the temporal sampling frequency for synthetic observations in the assimilation was matched to this daily resolution to enable future comparisons with APP-x. However, it should be noted that other operational products, such as the CLARA-A3 dataset from the EUMETSAT CM SAF, provide SIAL estimates at a pentad (5-day) resolution. This coarser temporal frequency reflects the need to combine observations in order to mitigate the effects of low solar elevation and oblique satellite viewing angles, which amplify bidirectional reflectance distribution function (BRDF) related uncertainties and complicate the feasibility of truly daily SIAL retrievals, particularly in polar regions (Riihelä et al., 2024).



While daily assimilation of variables like SIAL and SIT is useful for idealized benchmarking, assuming the availability of fully gridded daily observations may not align with current satellite capabilities. SIAL retrievals depend on favorable surface illumination and low cloud cover, while SIT estimates–though available along satellite tracks at daily resolution–require radar or lidar altimetry, which have limited spatial coverage and higher uncertainty during melt conditions or over thin ice (Karlsson et al., 2023; Petty et al., 2023). Future work should explore the impacts of more realistic observational sampling frequencies to

better reflect operational constraints.

## 2.4   Error Metrics

The primary metric used to evaluate assimilation performance is the Root Mean Square Error (RMSE), chosen for its robustness and interpretability. RMSE is particularly effective because it is sensitive to large errors, thereby highlighting substantial discrepancies between $\mu$ and the designated TRUTH member. It has also been widely adopted in previous sea ice DA studies

(Williams et al., 2023; Zhang et al., 2021). Additionally, RMSE yields a single scalar value that captures the overall magnitude of error, facilitating direct comparisons across different assimilation configurations.

For completeness, the mean absolute error (MAE) was also calculated, which yielded results that were qualitatively similar to those from RMSE (not shown). However, RMSE is prioritized here because large deviations, specifically, cases where $\mu$ strays $\geq 2\sigma$ from the synthetic observational TRUTH, can indicate observation rejection. This effectively results in a secondary

ensemble free-run, which may diverge even further from the TRUTH than the control simulation.

Importantly, RMSE accounts for both systematic errors (bias) and random errors (variance), offering a comprehensive measure of model performance. This holistic evaluation ensures that consistent and unpredictable errors are reflected in the metric. The RMSE is computed as follows:

$$\text{RMSE} = \sqrt{\frac{1}{N} \sum_{i=1}^{N} (y_i - \hat{y}_i)^2}, \tag{1}$$

where $N$ is the total number of data points, $y_i$ is the observed value at time $i$, and $\hat{y}_i$ is the corresponding predicted or modeled value.

To assess the robustness of assimilation performance, RMSE was computed across ten distinct ensemble members, each randomly selected to serve as a synthetic TRUTH in separate experiments. These ten TRUTH realizations were drawn from the same prior ensemble used in the assimilation cycles, ensuring internal consistency. By evaluating RMSE across multiple

TRUTHs, we account for natural variability in the system and avoid overfitting our results to a single realization. The reported RMSE values therefore reflect the mean performance across these ten cases, with 95% confidence intervals derived via bootstrap resampling. This multi-TRUTH framework supports a more generalized evaluation of each assimilation configuration.

To evaluate the relative benefit of assimilating SIAL observations compared to SIC or SIT, we additionally compute a percent RMSE difference metric. This metric quantifies how much more (or less) effective SIAL assimilation is at reducing RMSE

relative to SIC or SIT assimilation, with all differences referenced to a free-running control simulation (Figure 2). The metric is defined as:





$$\Delta\% \text{ Diff}_{\text{SIC, SIT}} = \left( \frac{\text{RMSE}_{\text{SIAL}} - \text{RMSE}_{\text{free run}}}{|\text{RMSE}_{\text{free run}}|} \right) \times 100$$
$$- \left( \frac{\text{RMSE}_{\text{SIC, SIT}} - \text{RMSE}_{\text{free run}}}{|\text{RMSE}_{\text{free run}}|} \right) \times 100 \tag{2}$$

A negative value of this metric indicates that SIAL assimilation resulted in greater RMSE reductions than SIC or SIT assimilation, whereas a positive value indicates poorer performance. This approach allows a direct comparison of the added value of SIAL assimilation across experiments, observational uncertainty levels, and regions.

## 3 Icepack Variable Relationships

We begin by examining the relationships among all `Icepack` aggregate variables of interest to assess their uniqueness and find where there is overlap (Figure 3). We find that many variables are strongly correlated with each other. In particular, SIAL is well correlated with SIC but not with SIT.

Due to the high correlation among the narrow-band components of SIAL in `Icepack`, we have consolidated these components into a single broadband SIAL for analysis. This simplification enables direct comparison of SIAL with SIC and SIT without reducing the overall complexity of the SIAL output. While all four narrow-band components are assimilated within the DA framework, our analysis focuses on this derived broadband component of the model output, defined as

$$\alpha_{\text{broadband}}(SIAL) = (0.00318 \times \alpha_{\text{DirVis}} + 0.00182 \times \alpha_{\text{DirIR}}$$
$$+ 0.63282 \times \alpha_{\text{IndVis}} + 0.36218 \times \alpha_{\text{IndIR}}). \tag{3}$$

From this point on, all references to "SIAL" refer to the broadband albedo component. The weighting for each of the broadband components is provided in the CICE Consortium (2025).

## 4 Results

### 4.1 RMSE Calculation and Statistical Significance

We calculate the ensemble mean ($\mu$) RMSEs for SIC and SIT relative to ten distinct TRUTH simulations, each based on a different randomly selected ensemble TRUTH member, for multiple assimilation experiments: SIC-only, SIAL-only (at low, medium, and high uncertainty levels; see the *Methods* section), SIT-only, all-variable assimilation (with medium SIAL uncertainty), and a control free-run (Figure 4). Error bars represent 95% confidence intervals around the mean RMSE, calculated across 10 TRUTH simulations (ensemble members 3, 5, 10, 12, 14, 16, 19, 21, 25, and 28) using bootstrap resampling.

Figure 5 summarizes which 2011 assimilation experiments result in significantly different RMSE (relative to the TRUTH) outcomes ($p < 0.05$), grouped by region and RMSE type (SIC or SIT). For each pair of experiments, we first apply the Shapiro–Wilk test to assess the normality of the RMSE distributions. If both distributions are approximately normal, we use



**Figure 3.** Correlation matrix ($R$) of `Icepack` main aggregate variables of interest averaged across all four regions specified in this study. Note that $h_{snow}$ represents the snow depth averaged across the grid cell.

Welch's $t$-test; otherwise, we employ the non-parametric paired Wilcoxon test. In the upper triangular matrix of each panel,

arrows denote statistically significant differences, pointing from the higher to the lower mean RMSE relative to the TRUTH. Arrow thickness reflects the strength of the statistical evidence, with thicker arrows corresponding to smaller $p$-values. Comparisons that do not meet the significance threshold are labeled as "ns" (not significant). We provide the full set of $p$-values in Supplement Table S1. We obtain similar results when analyzing other free-run years, suggesting that the findings are not year-specific (not shown).





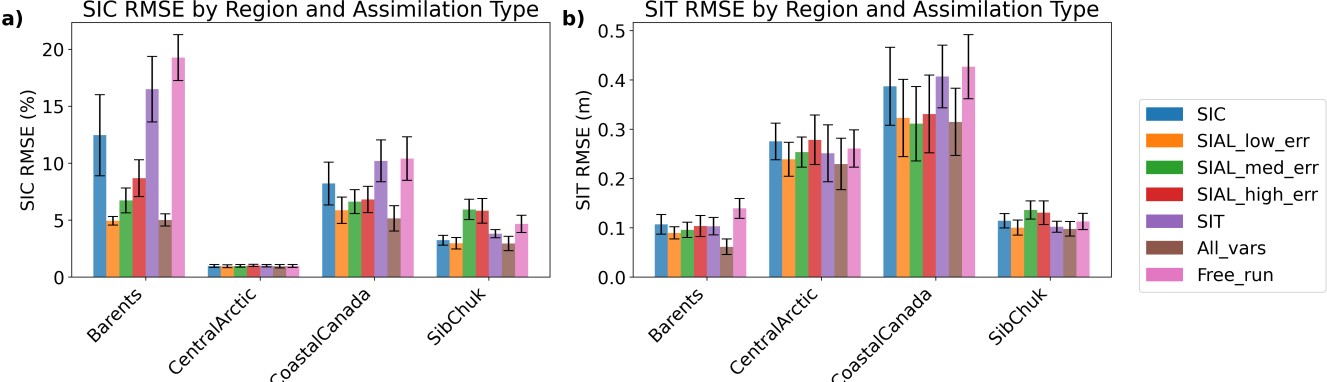

**Figure 4.** Panels (a) and (b) show the ensemble mean RMSEs relative to the TRUTH for 2011 SIC and SIT, respectively, across four key Arctic regions (Barents Sea, Central Arctic, Coastal Canada, and Siberian-Chukchi Sea), averaged over 10 TRUTH simulations. Error bars represent 95% confidence intervals computed via bootstrap resampling across the 10 TRUTH cases.

The statistical significance of the improvements in RMSE (relative to the TRUTH) varies substantially by region (Figure 5). As expected, no statistically significant differences are found in SIC or SIT RMSE within the Central Arctic. This is attributable to the presence of thick perennial ice with consistently high SIC, resulting in uniformly low SIC RMSE and high SIT RMSE that are not easily reduced by DA due to our prescribed SIT uncertainty (see Table 1).

We observe the largest RMSE improvements relative to the TRUTH within the Barents Sea region. In particular, there is

no statistically significant difference in SIC RMSE between assimilating only SIAL (with low observational uncertainty) and assimilating all variables. However, this equivalence breaks down when the SIAL observational uncertainty increases; medium and high uncertainty cases yield significantly different outcomes. Additionally, SIAL assimilation–under both low and medium uncertainty–significantly outperform SIC assimilation. Across all uncertainty levels, SIAL assimilation also outperforms SIT assimilation and the free run in terms of SIC RMSE, and is on par with SIT assimilation in terms of SIT RMSE. Regardless

of the assumed observational uncertainty, SIAL assimilation consistently improved the sea ice forecast relative to the free run–an improvement that we do not observe with SIT assimilation. SIAL assimilation also significantly reduces SIT RMSE in the Barents region compared to the free run. Notably, for SIT RMSE, assimilating any single variable performed statistically worse than assimilating all variables (assuming medium SIAL uncertainty).

In Coastal Canada, SIT RMSE results relative to the TRUTH are largely insignificant, except that SIAL assimilation per-

forms better than the free run, unlike SIC or SIT assimilation. For SIC RMSE, SIAL assimilation–regardless of observational uncertainty–significantly outperforms the free run. This is not the case for SIT assimilation alone, which shows no statistically significant improvement over the free run.

Despite the consistent benefits of SIAL assimilation in the Barents and Coastal Canada regions, results from the Siberian-Chukchi Sea are more nuanced. For SIC RMSE relative to the TRUTH, the observational uncertainty associated with SIAL

plays a critical role. Only SIAL assimilation with low observational uncertainty produces statistically significant improvements





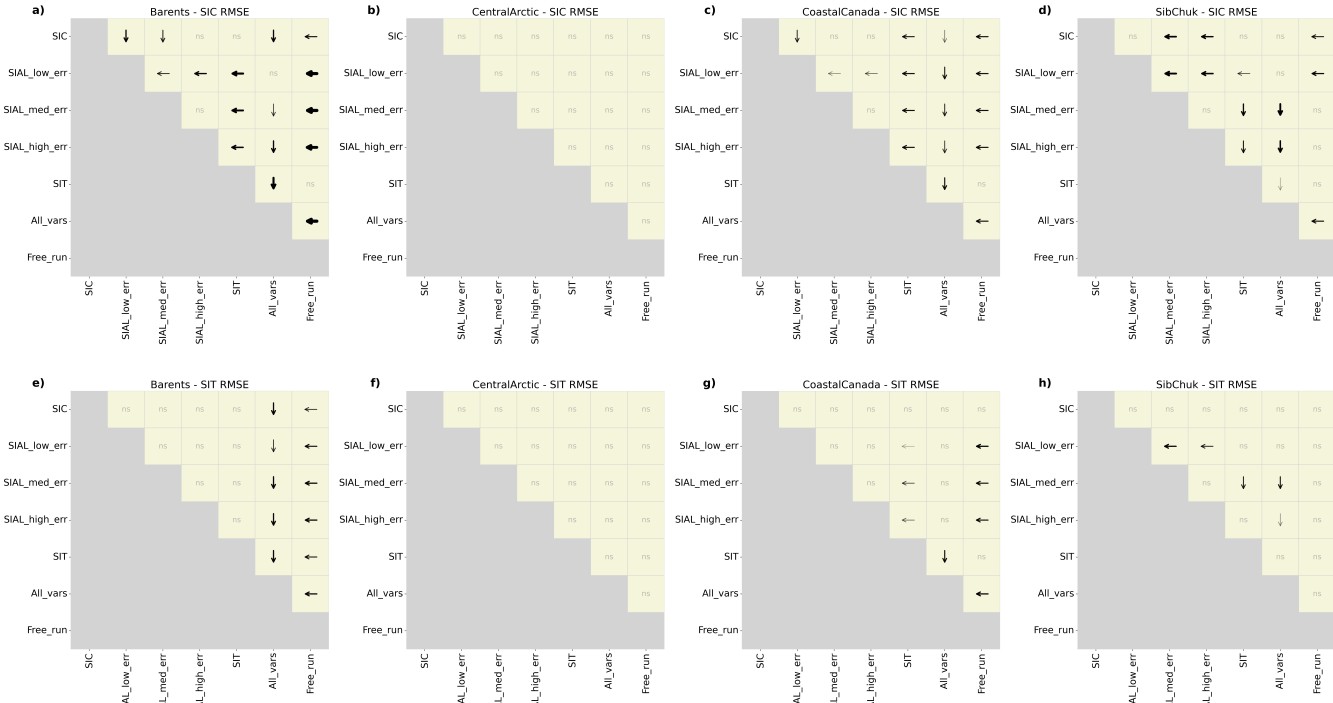

**Figure 5.** Pairwise statistical comparison of RMSEs (relative to the TRUTH) for different assimilation experiments across four regions for SIC (a–d) and SIT (e–h). Arrows indicate statistically significant differences ($p < 0.05$), pointing toward the assimilation experiment with lower RMSE; arrow thickness increases with statistical confidence (smaller $p$-values). The label "ns" denotes non-significant comparisons. As an example, the top left tan square of panel (a) compares SIC assimilation with low-uncertainty SIAL assimilation in the Barents Sea. The arrow points toward SIAL, indicating that low-uncertainty SIAL assimilation yields significantly lower SIC RMSE compared to the TRUTH than SIC assimilation alone. Assimilation cases include SIC, SIAL (low, medium, and high uncertainty), SIT, all variables (with medium SIAL uncertainty), and a free-run control. Exact $p$-values are provided in Table S1.

over the free run and SIT assimilation. When the SIAL uncertainty cannot be constrained to a low value, SIC assimilation statistically outperforms SIAL assimilation in this region. This finding underscores the importance of accurate SIAL uncertainty quantification to ensure robust and meaningful assimilation outcomes. Figure 4 further illustrates that the ensemble mean SIC and SIT RMSEs in the Siberian–Chukchi region are higher for SIAL assimilation with medium or high uncertainty than even the free run.

## 4.2 Percent Difference Metrics to Compare Assimilation Experiments

Figure 6 shows that the added value of SIAL assimilation over SIC or SIT assimilation varies substantially by region and uncertainty level. In general, low-uncertainty SIAL assimilation often outperforms SIC or SIT assimilation (blue shading), particularly in regions with intermediate ice conditions, while high-uncertainty SIAL assimilation frequently results in worse





performance (red shading). This underscores the importance of accurately characterizing observational uncertainty when in-
corporating albedo information into sea ice assimilation.

Figure 6 also provides insight into the internal consistency of each assimilation experiment across ensemble members.
Within each panel, the horizontal arrangement of ensemble TRUTH members (along the x-axis) allows for direct comparison
of RMSE outcomes under different regions and uncertainty levels. When a given row (i.e., region) is dominated by a single
color or shade, this indicates that the impact of SIAL assimilation relative to SIC or SIT assimilation is consistent across the
ensemble. For example, the Barents and Coastal Canada regions exhibit predominantly blue shading for SIC RMSE under low
SIAL observational uncertainty, suggesting robust improvements from SIAL assimilation relative to SIC or SIT assimilation
across nearly all ensemble members. In contrast, rows with a mixture of red and blue shading–such as the Central Arctic
and Siberian–Chukchi regions under medium and high SIAL uncertainty–reflect less agreement among ensemble members,
implying greater sensitivity to initial conditions or synthetic observational noise.

This spatial and ensemble-level agreement underscores the reliability of conclusions drawn from regions with consistent
shading and highlights the importance of ensemble spread in evaluating assimilation performance. Notably, trends appear to
be more consistent across regions for SIC RMSE than for SIT RMSE when comparing SIAL to either SIC or SIT assimilation.
This observation is consistent with results shown in Figure 5, where few single-variable assimilation experiments (e.g., SIAL
with low observational uncertainty) statistically outperformed others for SIT RMSE.

### 4.3    Uncovering Model Deficiencies via Category-Wise SIAL Assimilation in the Siberian–Chukchi Sea

From Figures 5-6, it is evident that SIAL assimilation, when not constrained to a low observational uncertainty, performs
poorly compared to other commonly assimilated sea ice variables within the Siberian-Chukchi Sea. To further investigate these
disadvantages, category-wise DA was conducted specifically for SIAL in this region. Category-wise DA refers to the assimila-
tion of a variable independently within each category of the ice thickness distribution represented in Icepack. Although this
approach is not applicable in real-world settings–since current observational systems cannot resolve sub-grid-scale ITDs–it is
particularly valuable in perfect model experiments. Such an approach helps identify underlying causes for unexpected assimi-
lation outcomes, such as cases where assimilation leads to an increase in ensemble mean RMSE relative to the TRUTH in our
model configuration.

Figure 7 illustrates the impact of assimilating category-specific SIAL observations at the medium uncertainty level ($\pm14\%$).
To represent uncertainty in broadband albedo as a function of ice thickness category, we applied a skewed uncertainty dis-
tribution in which thinner ice categories were assigned greater albedo uncertainty than thicker categories. This reflects the
assumption that observational and modeling challenges in characterizing surface conditions (e.g., open water, melt ponds) are
more prevalent over thin or newly formed ice (Nicolaus et al., 2012; Perovich et al., 2002). While this remains a broad approx-
imation, the objective of this assimilation experiment is not to rigorously quantify category-wise SIAL uncertainty, but rather
to diagnose potential deficiencies in the model's aggregate assimilation behavior.





Each thickness category was associated with a representative albedo range, and uncertainties were distributed using a physically informed, inverse-weighted scheme:

$$\sigma_n = \sqrt{\left(\frac{w_n \cdot \varepsilon_{\text{total}}}{\sum w_n}\right) \cdot \frac{(\Delta a_n)^2}{12}}, \qquad (4)$$

where $\sigma_n$ is the standard deviation of albedo assigned to the $n^{\text{th}}$ thickness category, $\Delta a_n$ is the width of the representative albedo interval, and $w_n = 1/(a_{n,\text{right}} + 1)$ is an inverse function of the upper bound of the interval. This weighting scheme places greater uncertainty on low-albedo, thin-ice categories. The quantity $\varepsilon_{\text{total}} = 0.14^2$ represents the total variance implied by a non-Guassian $2\sigma$ range of 14% (i.e., $2\sigma_{\text{total}} = 0.14$). Division by 12 assumes a uniform distribution of uncertainty within each albedo bin. This formulation concentrates uncertainty in the lower-albedo, thin-ice regime, where albedo retrievals and
surface classification are likely least reliable.

Applying this category-specific assimilation results in significant improvements in reducing SIC and SIT RMSE within the Siberian-Chukchi Sea compared to most aggregate assimilation experiments. While this methodology was also applied to other regions, results there were either insignificant or showed worsened performance, motivating a focused analysis on the Siberian-Chukchi region for this portion of the study. The top row of Figure 7 demonstrates that assimilating SIAL by category–assuming
the medium skewed uncertainty distribution described in Eq. 4–outperformed all other assimilation strategies for SIT RMSE, including experiments that assimilated all aggregate variables. For SIC RMSE, SIAL by category assimilation also performed on par with, or better than, all other assimilation configurations.

What causes this significant RMSE improvement relative to the TRUTH that is missed in the aggregate synthetic observations? The bottom-left panel of Figure 7 offers some insight. This plot shows the SIAL by category, averaged over ten
different ensemble TRUTH members. Notably, SIAL evolves differently across categories: in particular, SIAL in the thinnest ice category ($n = 1$) decreases rapidly in early April, while the remaining categories exhibit relatively stagnant behavior. This decoupled, category-specific SIAL trend is unique to the Siberian-Chukchi Sea and is not observed in other regions, helping explain why aggregate SIAL assimilation generally performs well or comparably in other areas—even under higher observational uncertainty. In contrast, within the Siberian-Chukchi Sea, `Icepack` does not evenly distribute the decrease in SIAL
across thickness categories. This introduces significant deficiencies in aggregate assimilation when SIAL uncertainty is unconstrained. In effect, the aggregate SIAL diverges from the true category-level dynamics, leading to an improper DA adjustment of model state variables $aice_n$ and $vice_n$. When uncertainty is high, this results in larger SIC and SIT RMSE relative to the TRUTH than even the free-running control simulation (see Figures 4-5).

The issue originates within `Icepack`: as $SIC_{n+1}$ decreases and $SIC_n$ increases, $SIAL_{n+1}$ decreases, but $SIAL_n$ does
not increase correspondingly to account for the influx of ice transitioning from thicker to thinner categories (see bottom row of Figure 7). This SIAL adjustment is likely (partially) incorrect and influences the formation of melt ponds within the thermodynamics module. Recent developments in CICE have aimed to improve the parameterization of the classification of the level ice melt pond; however, as of the time of this publication, these improvements have not yet been incorporated into operational `Icepack`. We acknowledge that aggregate SIAL observations in the real world can further complicate this discrepancy–





$SIAL_n$ may behave differently across categories in the model, and such a variation would not be accurately updated using a coarse aggregate SIAL observation.

## 5  Discussion

### 5.1  Mechanisms for RMSE Reduction and Benefits of SIAL Assimilation

As highlighted above, SIAL DA represents a novel approach with numerous benefits to improve sea ice modeling. Currently,
SIT is widely considered a robust assimilation variable due to its high sensitivity to physical processes, long memory, and predictive power. For instance, studies have shown that assimilating SIT (especially in winter) in models significantly enhances the accuracy of sea ice extent forecasts, underscoring its predictive value (Song et al., 2024; Williams et al., 2023; Ono et al., 2020).

We propose three mechanisms to explain how real-world SIAL aggregate assimilation may complement or replace SIT
aggregate assimilation and contribute to the observed reduction in RMSE across SIC and SIT (Figs. 4-5), which are directly related to model state variables:

1. **SIAL Observational Uncertainty vs. SIC and SIT Uncertainties.** Relative to SIC and especially SIT, SIAL observational uncertainty is likely lower, particularly during the summer months. The known uncertainty for SIAL is between 0-14% (Karlsson et al., 2023). In contrast, SIC known observational uncertainty is likely on the order of 5-10% (Zhang
et al., 2021; Peng et al., 2013), but higher in the MIZ. SIT observational uncertainty varies, ranging from 10-100% depending on ice thickness. Therefore, during summer months when the ice is thin, the observational uncertainty may approach or even exceed the SIT measurement (Song et al., 2024). Note that we assumed an optimistic SIT uncertainty of ±10% in this study. The actual uncertainty is likely higher, especially when using older observational data such as from CryoSat-2 (Chen et al., 2024). This suggests that SIAL and not SIT assimilation during summer months may be
beneficial, as SIAL leads to more constrained sea ice simulations when SIT observations are particularly sparse and uncertain.

2. **SIAL Assimilation as an Early Melt Onset Indicator.** SIAL assimilation provides an early indication of melt onset, which can mitigate low inflation during the early melt season. SIAL, acting as a proxy for snow cover and melt ponds, often exhibits a gradual decrease before SIC begins its rapid decline in the summer months. This early signal helps con-
strain the bounded assimilation of state variables $aice_n$ and $vice_n$, effectively moderating inflation in the model during late summer. The mean incremental changes in SIAL data assimilation (with medium uncertainty) are substantially larger during the early melt season compared to those from SIC or SIT assimilation, often resulting in improved agreement with the TRUTH member in early summer (Figure A2).

Careful monitoring of the adaptive inflation scheme is crucial, as SIAL, SIC, and/or SIT assimilation may result in
insufficient model spread, particularly when observations are assigned a low observational uncertainty. Xiong et al. (2002) emphasized the importance of balancing observational uncertainty assignments to avoid introducing excessive



bias or overconfidence in assimilation outcomes. This balance is especially significant during the early melt season, where reduced spread from sea ice DA can impact the model's responsiveness to rapidly changing ice conditions. See the supplementary information for additional detail about the adaptive inflation used in this study.

3. **SIAL Provides Contextual Clues for Sea Ice Fractional Coverage.** SIAL is a critical variable in understanding the fractional coverage of different surface types within a grid cell, such as snow-covered ice, bare ice, and melting ice. The assimilation of SIAL provides continuous insights into the energy balance of the sea ice surface, which is essential for accurately predicting seasonal changes in sea ice extent and thickness. Springtime melt ponds, for instance, are strong indicators of the September sea ice minimum (Schröder et al., 2014). Snow cover and late-season snowfall also influence

SIAL and, consequently, the rate of ice melt (Chapman-Dutton and Webster, 2024; Vérin et al., 2022; Perovich et al., 2017)

In `Icepack`, SIAL can be expressed as a weighted sum of the contributions from snow, melting ice, and bare ice surfaces:

$$\alpha_{\text{broadband}} = f_{\text{snow}}\alpha_{\text{snow}}(T, \text{age}) + f_{\text{melt}}\alpha_{\text{melt}}(T, h_{\text{melt}})$$

$$+ f_{\text{ice}}\alpha_{\text{ice}}(T, h_{\text{ice}}) \tag{5}$$

where:

- $f_{snow}$, $f_{melt}$, and $f_{ice}$ are the fractional area weights for snow-covered ice, melt ponds, and bare ice, respectively,

- $\alpha_{snow}$, $\alpha_{melt}$, and $\alpha_{ice}$ are the albedo values for each surface type,

- T represents temperature,

- $age$ is the snow age,

- $h_{melt}$ and $h_{ice}$ are the melt pond depth and ice thickness, respectively.

This formulation highlights how SIAL serves as a diagnostic variable that encapsulates the physical processes driving seasonal surface transitions in the Arctic. As we have seen from our results, by assimilating SIAL, the model is better equipped to constrain observed variables like SIC and SIT, as SIAL (aggregate) acts as an integrative measure of surface

state changes during the melt season. This contextual information has been shown above to help reduce biases and improve forecasts of Arctic sea ice behavior.

SIAL is more than a simple proxy for fractional coverage; it encapsulates key information about the radiative and thermodynamic state of the sea ice surface. By capturing both spatial and spectral variations in ice and snow reflectivity, SIAL helps constrain the energy balance of the ice–ocean–atmosphere system and offers insight into melt processes, pond evolution, and

snow metamorphism. This richness of information allows SIAL to provide a form of memory for the ice system–similar to



ice age, which has been shown to be a robust indicator of sea ice state (Zhang et al., 2018). However, unlike ice age, which may lose relevance as the Arctic transitions to a primarily first-year ice regime (Sumata et al., 2023; Meier et al., 2023), SIAL is expected to remain an important variable. Its demonstrated utility in highly seasonal regions like the marginal ice zone (e.g., Barents Sea) underscores its value, and its importance may further grow as Arctic precipitation shifts from snow to rain

(McCrystall et al., 2021), altering surface albedo and melt dynamics in ways that SIAL can continue to capture.

## 6  Conclusions

The results of this study suggest that SIAL assimilation is a robust alternative to assimilating traditional sea ice variables, such as SIC and SIT. SIAL assimilation performs well due to its relatively low observational uncertainty, its ability to better predict the melt onset, and its intrinsic relationship to snow, bare ice, and melt pond fractional coverage. In this one-dimensional

`Icepack` experiment, SIAL assimilation performs on par with or significantly better than SIC and SIT assimilation in all studied regions except the Siberian-Chukchi Sea. Within the Siberian-Chukchi Sea, SIAL assimilation performs well assuming low uncertainty, but model physics and higher observational uncertainty pushes SIAL assimilation to perform worse than SIC and SIT assimilation.

We acknowledge that this study was conducted in a one-dimensional idealized environment and emphasize the need for

further experimentation using a fully coupled global climate model to comprehensively assess the impact of SIAL assimilation on the mean sea ice state in a multi-dimensional framework. In particular, future research should investigate the performance of satellite-derived SIAL assimilation within a three-dimensional QCEFF to better quantify the advantages and limitations identified in this study.

A key challenge is the limited understanding and quantification of uncertainties associated with satellite SIAL retrievals.

This remains an underexplored area, and our study encountered difficulties in establishing a consensus on how best to represent these uncertainties. We advocate for additional field campaigns that measure SIAL across different spectral bands and seasons to support more robust cross-validation of satellite products. A long-term, in-situ observational record is also essential to better constrain satellite SIAL uncertainty estimates, which would enhance their utility in data assimilation frameworks.

Despite these challenges, we remain optimistic about the potential of SIAL assimilation, particularly given the extensive

availability of high-resolution satellite SIAL products spanning the satellite era. Notably, the CLARA-A3 surface albedo product, produced by EUMETSAT, provides pentad SIAL estimates at 25 km resolution on the Equal-Area Scalable Earth (EASE) grid across multiple narrow-band frameworks (Vid. Supp. 1). Similarly, APP-x, developed in collaboration with NOAA and the University of Wisconsin, offers twice-daily narrow- and broadband SIAL composites at the same resolution (Vid. Supp. 2). These observational datasets offer a promising alternative to conventional SIC and SIT assimilation approaches and hold

considerable potential for improving sea ice reconstruction over the satellite observational period (1982–present). Continued investigation of these products and their incorporation into advanced data assimilation systems is critical for unlocking their full potential.



*Code availability.* https://github.com/jrotondo-uw/cice-scm-albedo-da. The repository contains all scripts, model modifications, and Jupyter notebooks required to reproduce the assimilation experiments and final figures. The workflow includes preprocessing, data assimilation using

DART with `Icepack`, and post-processing for analysis and visualization. Instructions for modifying DART and `Icepack` to assimilate sea ice albedo are also provided.

*Data availability.* The final post-processed NetCDF files used to generate the figures in this study are available at: https://doi.org/10.5281/ zenodo.15571204. The raw model free runs, synthetic observations, and intermediate data are not included due to size constraints but can be reproduced by following the workflow described in the *Code Availability* section.

*Video supplement.* https://github.com/jrotondo-uw/cice-scm-albedo-da/tree/main/Video_Supplements

**Appendix A**

**A1**

*Author contributions.* JR led the project, performed the data assimilation experiments, and conducted the analysis. MW contributed significantly to model development and implementation. CB supervised the project and provided ongoing guidance throughout. RC and SC offered

constructive feedback and oversight during the development and interpretation phases. All authors contributed to the final manuscript.

*Competing interests.* The authors declare that they have no conflict of interest.

*Acknowledgements.* We thank several people for helpful conversations, such as Walt Meier, Edward Wrigglesworth-Blanchard, Gregory Hakim, and Jeffery Anderson. This work was supported by the National Science Foundation under Awards PLR-2141538 and PLR-1936428, the American Meteorological Society Graduate Fellowship supported by the National Oceanic and Atmospheric Administration (NOAA)

Climate Program Office (CPO), and from NASA ROSES Grant Number 80NSSC21K074.



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





**Figure 6.** Percent RMSE differences between SIAL assimilation and either SIC or SIT assimilation across Arctic regions. Each column shows SIAL assimilation results under low, medium, and high observational uncertainty (left to right). Rows are grouped by RMSE metric relative to the TRUTH and comparison type: SIC RMSE under SIC (a–c) and SIT (d–f) assimilation, and SIT RMSE under SIC (g–i) and SIT (j–l) assimilation. Values are computed relative to the free-running control using Equation 2. Each cell shows the difference between the RMSE percent reduction achieved by SIAL assimilation and that of SIC or SIT assimilation; negative values (blue) indicate better performance by SIAL assimilation, while positive values (red) indicate worse performance. Darker shading reflects greater magnitude. Values are shown per region and ensemble TRUTH member.



**Figure 7.** Siberian-Chukchi Sea results for the SIAL category-wise assimilation experiment. Top-left (a): Pairwise statistical significance of differences in mean RMSE for SIC among assimilation configurations. Black arrows indicate statistically significant differences in RMSE between assimilation configurations ($p < 0.05$), pointing toward the experiment with lower RMSE relative to the TRUTH. "ns" denotes differences that are not statistically significant. Red boxes highlight comparisons between $SIAL_{category}$ and all other assimilated variables and uncertainties. Top-right (b): As in the top-left but for SIT RMSE. Bottom-left (c): Time series of domain-averaged broadband albedo ($\alpha_{broadband}$) for various SIAL ensemble category members (n = 1 to 5). Bottom-right (d): Corresponding time series of SIC (%) for the same ensemble members, including TRUTH, free run, and analysis under category members (n = 1 to 5).



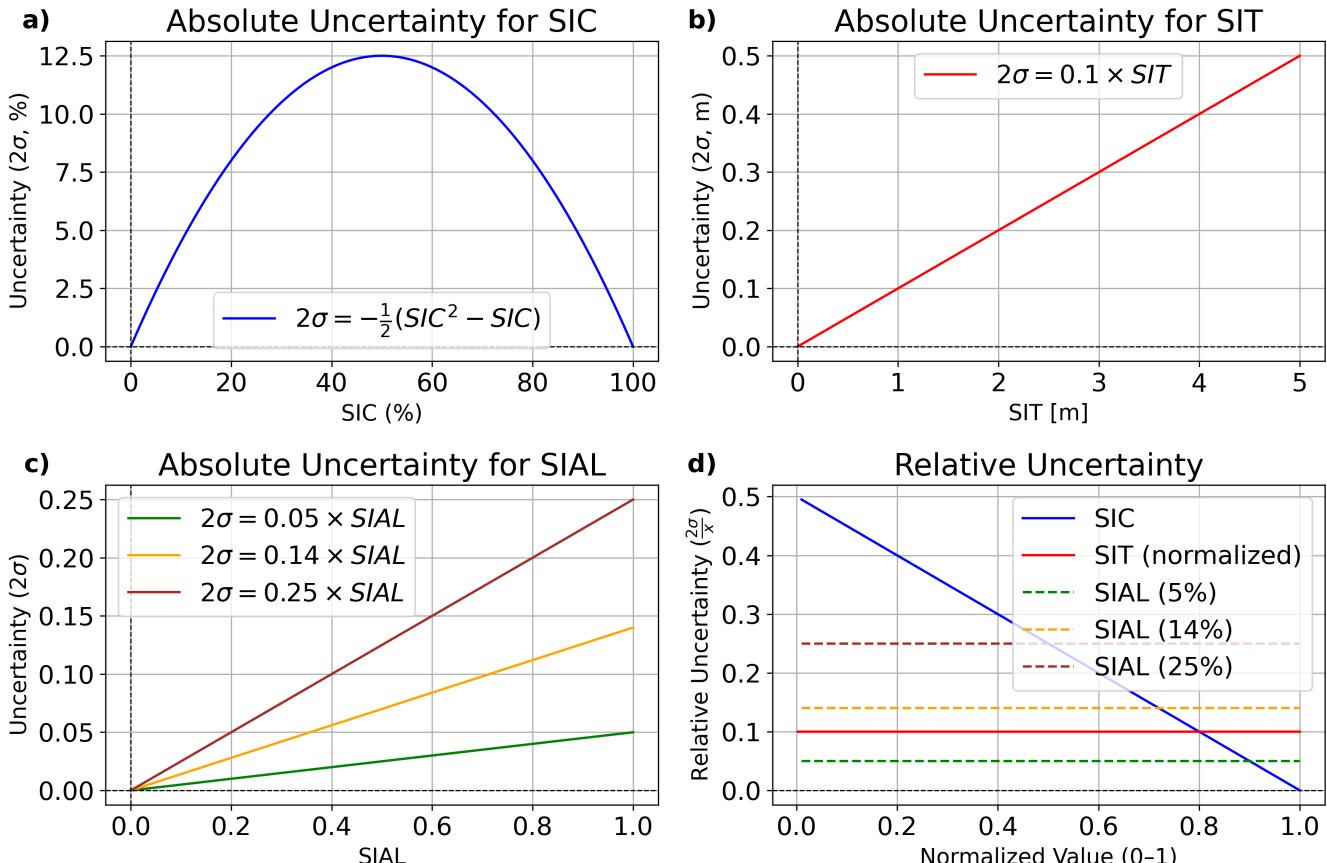

**Figure A1.** Observational uncertainty distributions for SIC (a), SIT (b), and SIAL (c). Note that there are three different uncertainty scenarios for SIAL to account for unknowns in SIAL retrieval estimates. The normalized amount of uncertainty by variable (0-1) is included in the bottom-right (d) for comparison.





**Figure A2.** Mean magnitude of DA increments in SIC, expressed as absolute percent change from the prior state, averaged across 10 assimilation experiments. Each panel corresponds to a different assimilation configuration: SIAL (a; medium uncertainty), SIC (b), and SIT (c). Data are smoothed using a 7-day rolling average. Regions with rare high-magnitude increments (>10%) are annotated with their maximum value for clarity. Notably, the SIAL DA configuration shows pronounced early-season activity, particularly in the Barents Sea region, while late-season peaks in Coastal Canada are prominent in all configurations.