# Peer review of "Sea Ice Albedo Bounded Data Assimilation and Its Impact on Modeling: A Regional Approach"

_EGUsphere, 2025_

## Referee Comment (RC1)

Review of "Sea ice albedo bounded data assimilation and its impact on modelling: A regional approach"

By Joseph F. Rotondo et al.

Version 1

Date of review: 10 September 2025

**Summary**

This manuscript presents an interesting study on the potential usefulness of assimilating sea-ice albedo observations in sea-ice modelling and predictions. Although the study is restricted to a single-column sea-ice model, its promising results merit future investigations on this topic in the context of a full, three-dimensional model. The reporting of findings from the present study is therefore highly relevant to the scientific community. Hence, noting that the material in the manuscript is within the remit of *The Cryosphere*, I do recommend the manuscript's acceptance in the journal, though subject to some minor corrections as detailed below.

**Main Comment**

In Section 4, you analyse the results from a number of experiments assimilating different types of observations or different combinations of observation types. These include SIC-only, SIT-only, SIAL-only, and all variables (SIC, SIT and SIAL) combined. As the paper is aimed at demonstrating the added value of assimilating SIAL observations, it would be good to run an SIC-and-SIT-only experiment and compare it with the all-variable experiment to single out the impact of additionally assimilating SIAL observations. The findings from this comparison can complement the discussion around the comparison between an SIAL-only experiment and a free run (which also demonstrates the added value of assimilating SIAL observations, but without the SIC and SIT constraints).

On a related note, in Section 5.1 you say that assimilating SIAL may "complement or replace" the assimilation of SIT (line 374). These two observation types provide complementary information about sea ice, so I can't see why one might want the assimilation of SIT to be replaced – as opposed to being complemented – by the assimilation of SIAL. To me, the focus should be on the added value of SIAL observations, rather than having only one or the other observation type (but not both).

**Other Comments**

1.  Lines 8 – 9: The use of "three-quarters of the Arctic regions studied" and "across all regions" could be potentially misleading, as you are only looking into 4 discrete points in the Arctic Ocean.
2.  Line 13: Is this a typo? "improved" → "improving"

3.  Lines 150 – 151: Does it mean there is no guarantee that the synthetic observations will be within the hard physical bounds?

4.  Lines 154 – 155: The phrase "as aggregates of modelled quantities categorized by thickness… over these thickness categories" sounds a bit clumsy and could be a bit confusing.  Is there a simpler way of expressing it?

5.  Line 165: Please elaborate on what is meant by "direct and indirect shortwave radiation are treated equivalently".

6.  Line 185: The acronym ICESat-2 is undefined.

7.  Table 1:
    a.  In the heading of the right-most column, what is the significance of the multiplier 2 in front of $\sigma$?  Why do you show uncertainties in the form of $2\sigma$ instead of $\sigma$?
    b.  I presume $aice_n$ and $vice_n$ are normalised quantities ("per unit area", i.e. normalised by the area of the grid cell); otherwise, the equations for the forward operators don't make sense.  This should be mentioned somewhere, ideally when $aice_n$ and $vice_n$ are first defined.
    c.  How often do you get SIC values of exactly 0 or 1?  In those cases, the observational uncertainty for SIC would be exactly 0.  Would this be theoretically problematic for the QCEFF?  Would you consider imposing a minimum value for the observational uncertainty?
    d.  In the expressions for the observational uncertainty of the different types of SIAL, the $\alpha$ should come with the appropriate subscripts.  It would be even better if you make it clear, ideally through different notations, that the observational uncertainties (across all observation types) are parametrised based on the **background** values of these model variables, as opposed to the (yet-to-be-known) analysis values.

8.  Lines 198 – 199: Does it mean that the assimilation window is 1 day long?

9.  Equation 1 and line 225: Is the summation in the equation taken over the data points of many time-instances?  How many data points are there?  Please clarify.

10. Line 227: "separately for" may be a better choice of words than "across".

11. Line 233: Do you mean you compare the SIAL experiments against SIC-only and SIT-only experiments?  (See also the Main Comment above.)

12. Equation 2: The subscript "SIC, SIT" could be misleading.  As things stand, the subscript refers to the reference experiment you compare against, but it is easy to misinterpret the subscript as the quantity over which the RMSE is computed (which is not indicated in the notations of Equation 2).  As you demonstrate in the rest of the article, these two aspects should be separate.

13. Lines 244 – 245: In Table 1 you show that SIAL is defined after normalisation by SIC.  With that in mind, what drives the finding here that SIAL is well-correlated with SIC?

14. Lines 272 – 273: You say that thick perennial ice in the Central Arctic results in "high SIT RMSE that are not easily reduced by DA due to our prescribed SIT uncertainty".  I am not sure how they are related (high observational uncertainty does not necessarily mean high RMSE).  What about in experiments that don't assimilate SIT observations?

15. Figures 6 and 7: Please move the figures up so that they could be close to the text that discusses about them.
16. Line 297: The phrase "the added value of SIAL assimilation over SIC or SIT assimilation" is not clear. Do you mean the comparison between the SIAL-only and SIC-only / SIT-only experiments, or do you mean the impact of additionally assimilating SIAL observations? (See also the Main Comment above.)
17. Line 319: The term "category-wise DA" isn't clear; you may consider using "assimilating category-specific observations" instead.
18. Equation 4 and lines 335 – 340: What are "the width of the representative albedo interval" ($\Delta a_n$) and the notation $a_{n,\text{right}}$? Also, you say that the formulation in Equation 4 "concentrates uncertainty in the low-albedo, thin-ice regime", but I don't find it easy to infer it from the equation. The motivation behind the equation needs to be more clearly explained.
19. Lines 350 – 351: That small $n$ refers to thin ice needs to be more explicitly mentioned somewhere.
20. Lines 352 – 353: "helping explain why aggregate SIAL assimilation generally performs well or comparably in other areas" – how?
21. Line 388: What does "inflation" mean in this context?
22. Lines 466 – 467: Is there an appendix missing?

---

## Author Comment (AC2)

Review: Sea Ice Albedo Bounded Data Assimilation and Its Impact on Modeling: A Regional Approach

This study investigates the impact of assimilating sea ice albedo (SIAL) on Arctic sea ice prediction using a series of perfect model experiments. The work is well-structured and addresses a relevant topic in data assimilation using the Quantile Conserving Ensemble Filtering Framework (QCEFF). The objective to evaluate the impact of SIAL assimilation on sea ice forecasts is clearly stated. The authors selected four diverse Arctic regions and use Icepack model for the assimilation with spin-up period (2000-2010) and experimental timeframe (2011-2015). The manuscript presents interesting results, particularly regarding the complementary benefits of multi-parameter assimilation. Several aspects of the experimental design and presentation could be clarified to strengthen the manuscript.

Thank you for sharing your overall favorable assessment.

**Specific comments**

1. Ensemble generation

The authors mentioned how to construct the ensemble but only in the abstract and L74 without details. It would be valuable to know the details on the ensemble generation, and elaborate in the Method section.

The authors appreciate this comment. The following section has been added to reflect this:

"To construct the ensemble, we employed the *Icepack* single-column sea ice model, configured to represent five Arctic regions of interest (Barents Sea, Coastal Canada, Siberian–Chukchi Seas, and the Central Arctic). Our ensemble was generated solely by perturbing external atmospheric forcing fields, following Appendix A of Wieringa and Bitz (in press), which ensures that perturbations retain realistic spatiotemporal coherence between atmospheric variables. In practice, this is achieved by perturbing the left singular vectors of the covariance matrix of the atmospheric state, such that the resulting ensemble members remain statistically consistent with the reference forcing fields.

Each ensemble member was initialized from a multi-year spin-up (2000–2010) to reduce sensitivity to initial conditions. Thirty ensemble members were produced for each experiment. For every member, a unique namelist file was created by perturbing selected atmospheric forcing fields (temperature, specific humidity, zonal and meridional wind, short- and longwave fluxes, and precipitation) using JRA-55-do reanalysis data. The ocean forcing remained identical across the ensemble. Specifically, the ocean component was represented by a slab ocean, with initial conditions extracted from the ocean output of a fully coupled historical CESM2 simulation. All ensemble members shared this common ocean forcing, ensuring that ensemble spread arose primarily from atmospheric forcing perturbations.

All members were integrated forward for the spin-up period using consistent external boundary conditions, after which restart files were generated at the end of 2010. These restart states served as the initial conditions for the assimilation experiments. Model output was collated across ensemble members, resampled to daily means, and combined into a single dataset with ensemble dimension. Ensemble mean and spread were calculated and stored in separate NetCDF files for subsequent analysis."

*Note:* The ocean initial conditions originate from a fully coupled CESM2 piControl simulation, corresponding to the B compset initialized in 1850 with the CAM6-WACM atmosphere. The prescribed ocean heat-flux forcing (qflux) is not taken directly from a single coupled simulation output, but is computed from climatological output of a fully coupled Ocean General Circulation Model (OGCM) following the methodology of Bitz et al. (2012). The resulting forcing file used here is pop_frc.b.e21.BW1850.f09_g17.CMIP6-piControl.001.190514.nc, previously archived on Derecho/Cheyenne (path may have changed over time). No additional appendix is provided, as this qflux forcing is applied directly at the model grid points used in this study.

2. Observation uncertainty

The assumptions for observation uncertainty are highly consequential for the DA results (Line ~170). The chosen setup—a parabolic distribution for SIC, 10% for SIT, and zero uncertainty at the bounds (0% and 100% SIC, 0m SIT)—is a common simplification but has significant implications. Setting uncertainty to zero forces the model to exactly match the observation at these bounds. This can lead to overconfidence and an artificial reduction in ensemble spread, potentially skewing the results. The authors could clarify this choice in the Methods or Discussion section.  A strong recommendation for future work would be to adopt a more realistic uncertainty that avoids zero uncertainty, for instance, by specifying a minimum uncertainty floor (e.g., 1-2%) for SIC at 0% and 100%.

The authors agree that a more realistic uncertainty floor should be implemented to avoid zero uncertainty. In the initial setup, a fail-safe minimum of $10^{-7}$ was employed to prevent DART from rejecting observations with zero uncertainty. However, as the reviewer correctly notes, such small values can artificially accelerate the collapse of ensemble spread and reduce realism.

To address this, we re-ran the experiments with revised observation uncertainty specifications: a minimum of 0.01 for SIC at its bounds (0 and 1) and a minimum of 0.02 for SIT at its lower bound (0 m). During testing, we also found that when SIC approaches ~0, Icepack automatically assigns aggregate SIAL a value of 0 to represent open ocean. To ensure consistency, we therefore introduced a minimum uncertainty of 0.01 for SIAL as well. These revisions provide a more realistic treatment of observation error at the boundaries, preventing overconfidence while maintaining numerical stability in DART.

Implementing these corrections led to notable changes in our results. Figures 4–7 and the accompanying text have been updated accordingly. The key outcome is that all assimilation experiments performed better under the revised uncertainty scheme, as it prevented the collapse of model spread. Assimilating all variables together still produced the best results, and Figure 6 has been updated to reflect the advantages of multi-variate assimilation over SIC-only and/or SIT-only assimilation. Thus, while the overall conclusions of the study remain unchanged, the individual significance tests between assimilation experiments differ slightly under the more realistic uncertainty floor.

3. Interpretation of SIAL vs. SIC assimilation results

The central conclusion that SIAL assimilation outperforms SIC assimilation under low uncertainty is compelling. However, this advantage may be partially confounded by the differential uncertainty settings applied to each variable. SIAL's assigned uncertainty is likely low (though not explicitly stated in the provided text), while SIC's uncertainty is structurally defined by a parabola, which is higher everywhere except the bounds. A fairer comparison would require testing SIC assimilation under similarly low uncertainty assumptions. The authors may state this potential confounding factor in the discussion (e.g., around Line 306).

We thank the reviewer for this thoughtful comment. We acknowledge that the differing uncertainty formulations for SIAL and SIC may influence the comparison between assimilation experiments. Our choice of uncertainty settings was guided by previous studies (e.g., Karellson et al., 2024 for SIAL; Wieringa et al., 2024 for SIC and SIT), which reflect the distinct observational retrieval methods underlying these variables and their associated uncertainties. Specifically, SIAL retrievals are based on radiative transfer, requiring knowledge of reflected shortwave radiative fluxes at the top of the atmosphere, whereas SIC retrievals rely on passive microwave signals that exploit the emissivity contrast between open ocean and sea ice. Because these retrieval techniques are fundamentally different, it would not be physically consistent to impose identical uncertainty structures for both variables.

That said, we agree with the reviewer that further work is needed to systematically assess the role of uncertainty assumptions in data assimilation (DA) performance. In particular, future studies could explore the sensitivity of SIC assimilation to alternative (e.g., lower or more uniform) uncertainty settings to better isolate variable-dependent effects. For SIAL, we emphasize that uncertainty characterization remains an open research question, and we plan to investigate this aspect further. As the reviewer alluded, SIAL uncertainty is likely low, but the lack of spatially comprehensive in-situ measurements makes it difficult to validate the satellite footprint.

To address the potential impact of uncertainty distribution on our results, we re-ran the SIC-only assimilation experiments assuming a maximum SIC error of 5%, consistent with the "low" uncertainty setting used for SIAL. The results, shown in the figure below (excluded from the

manuscript for brevity), indicate minimal impact of the maximum SIC error on DA performance. The only notable improvement occurs in the Barents Sea for SIC RMSE, yet it remains insufficient to replicate the performance of the SIAL-only assimilation under low error. This supports our argument, discussed in the manuscript, that SIAL serves as a better predictor of melt onset—helping prevent filter divergence and reducing SIC RMSEs for thin, seasonal ice. Altogether, these findings suggest that our results are driven more by the intrinsic characteristics of the assimilated variable and its covariance with model state variables (e.g., *aicen* and *vicen*) than by the exact uncertainty distribution. As an aside, we note that the uncertainties used for SIC and SIT in our study are both likely lower than those observed in reality (e.g., SIC: Zhang et al., 2022; SIT: Fiedler et al., 2022).

[Figure]

**Figure A:** Demonstration showing that varying the maximum SIC uncertainty (12.5% vs. 5%) has minimal influence on assimilation outcomes.

4. Multi-parameter assimilation

The finding that the simultaneous assimilation of SIC, SIT, and SIAL yields the best performance is a key result. The complementarity between these variables, as likely illustrated in Figure 4, is a highly valuable insight. This multi-parameter complementarity deserves to be highlighted and discussed in greater depth as a major takeaway of the study, perhaps exploring the physical reasons behind why the constraints provided by these variables are non-redundant.

The authors agree and thank the reviewer for this insightful comment. We have revised Figure 6 to emphasize the benefits of multivariate assimilation, focusing on the combined assimilation of SIC, SIT, and SIAL rather than individual comparisons of SIC-only, SIT-only, and SIAL-only experiments. This updated figure highlights the consistent advantages of assimilating multiple parameters over single-variable approaches. Accordingly, we are revising the discussion and conclusions sections to center on multi-variate assimilation as a primary theme.

The likely physical explanation for this improvement lies in albedo's role in capturing the melt season. As temperatures rise, *Icepack* simulates melt pond formation and decreasing snow cover, both of which provide additional information on ice melt processes. While SIC and ice volume may not yet show a steady decline, the reduction in snow and exposure of darker underlying surfaces allow albedo assimilation to precondition the ice to melt under higher model spread, despite albedo not being a prognostic variable. Consequently, when combined with SIC and SIT assimilation, the ensemble framework produces more physically consistent updates (particularly for SIC increments) leading to improved overall assimilation performance.

[Figure]

**Figure B:** Revised version of Paper Figure 6, emphasizing the benefits of multivariate assimilation when using the *level* melt pond scheme, except for Coastal Canada and the Siberian Chukchi Sea which use the *topographic* melt pond scheme (see below for more details on why we add this specification).

5.  Region-specific behavior of albedo (Section 4.3)

The identified unique albedo evolution for category `n=1` ice in the Chukchi Sea is interesting. Is there any reason for that? the formation of melt ponds? Why is it pronounced in this region and not in others?

We have delved deeply into this particular result, after developing several theories as to why SIAL assimilation in the Siberian-Chukchi Sea produced worse performance when error is unconstrained. First, we looked into the development of thinner ice in category $n = 1$ as suggested by the reviewer. Because *Icepack* only represents a single discrete point, the user must specify what to import when the ice undergoes closing. The available options are to import open water or uniform ice. For this region, we opted to import open water into the grid cell since this region is assumed to be near the sea ice edge most of the year. Hence, under closing, when existing ice is rafted or ridged, leads form. Upon refreezing thin, snow-free ice forms and populates category $n = 1$.

This behavior contrasts with that in the Coastal Canada region, which experiences a similar seasonal SIC cycle (as shown in Figure 2 of the main manuscript). There, we elected the uniform ice option, which imports ice with the same conditions (ice thickness distribution, snow depth, etc.) as are present in the grid cell as rafting and ridging occur. For clarity, a categorical comparison illustrating this behavior is provided in Figure C.

[Figure]

**Figure C:** The SIAL cycle per category $n$ between two regional ice coverage locations where Coastal Canada experienced uniform ice fluxing and Siberian–Chukchi Sea experienced open water fluxing.

In a fully dynamical ice configuration, the imported ice would likely lie between the open-water and uniform-ice closing cases. While this distinction may influence results to some extent elsewhere, it is unlikely to have a major effect, particularly at the satellite pixel scale. The chosen import scheme helps explain the degraded performance of SIAL assimilation relative to the free run in the Siberian–Chukchi Sea.

Because SIAL assimilation aggregates albedo information across categories, it introduces negative SIC innovations earlier in the melt season, leading to premature melt relative to the truth (as shown in the accompanying Figures D & E, excluded from the paper for brevity). From Figure 7, we see that assimilating SIAL by category avoids this early negative innovation. If

category-specific assimilation were implemented, the innovations would reveal that only SIAL in category $n = 1$ decreases substantially early in the Siberian–Chukchi Sea melt season, indicating that ice in other categories remains largely unaffected by melt. This information would constrain the assimilation to slow the rate of sea ice loss. In contrast, large negative aggregate innovations push SIAL states beyond the observational range, producing a model evolution that diverges from the truth and melts too early (Fig. E).

[Figure]

**Figure D:** Early melt-season negative innovations from SIAL assimilation in the Siberian–Chukchi Sea.

[Figure]

[Figure]

**Figure E:** New model state in the Siberian–Chukchi Sea showing reduced ice coverage—less than in the free run—resulting from negative innovations.

Since submitting the paper, **the authors have identified a more likely source of the discrepancy contributing to the higher-than–free-run error for SIAL assimilation in the Siberian–Chukchi Sea and the near-baseline performance in Coastal Canada.** The melt pond fraction in the version of *Icepack* (1.4.1) used for this study is unrealistically high compared to observations, which suggest a maximum melt pond coverage of roughly 50% during the melt season (Niehaus et al., 2024). As shown in Figure F below, the simulated melt pond fraction approaches unity during the melt season in Coastal Canada and the Siberian–Chukchi Seas, leading to erroneous sea ice updates. This unrealistic melt pond behavior is likely responsible for the early negative SIC innovations and associated negative updates discussed above. Below, we diagnose methods to assess the impact of melt pond schemes on our results, and suggest differential melt pond schemes for different regions.

Since this work was submitted, efforts have been underway to improve the melt pond parameterization in *Icepack*, with new schemes currently in development and submission. The authors have added this discussion to clarify why SIAL assimilation in the Coastal Canada and Siberian–Chukchi regions may not perform as well as initially anticipated within the 'Results' and 'Discussion' Sections. Notably, melt pond evolution appears more realistic in the Barents region (reaching only ~30% coverage before completely ice-free conditions), which likely explains why SIAL assimilation under the level melt pond scheme there produces markedly improved results.

To further investigate the source of this unrealistic melt pond evolution, the authors examined the melt pond scheme used in these experiments—the "level pond" scheme (see Icepack Documentation Section 2.7.2.2). In this scheme, ponds form and are evenly distributed across all level ice. Because closing rates are specified from the SHEBA campaign (1998), which took place in perennial sea ice conditions, they are likely biased low for seasonal ice locations (e.g., Coastal Canada and the Siberian–Chukchi Seas). As a result, the ice is too level (~95–100%, not shown), allowing melt ponds to quickly spread across the surface. During the melt season, this causes the simulated albedo to drop unrealistically, as if the surface were largely melt pond water. This mechanism likely explains the premature melt and excessive albedo reduction observed in the SIAL assimilation runs—a model representation error rather than a physical response. The authors therefore do not expect these early-melt artifacts to occur outside this idealized *Icepack* configuration. Incorporating full CICE, with its dynamic treatment of rafting and ridging, would likely yield a more realistic melt pond distribution in the early melt season and mitigate this issue.

[Figure]

**Figure E:** Erroneous melt pond behavior in the Coastal Canada and Siberian–Chukchi regions. Although melt pond depths are modest, the ponded areas appear excessive—unlike in the Barents Sea—leading to substantial negative SIC innovations and a new model state with reduced ice coverage.

Fortunately, the version of *Icepack* that we used for this study did have the ability to use the "topographic" melt pond scheme. This melt pond scheme is generally not recommended for general model use because it assumes an empirical form of melt ponds that does not transfer well to future climates based on conversations with Elizabeth Hunke, a main developer of the Community Ice CodE (CICE), which encompasses *Icepack*. For the purpose of our experiments, however, we reran all cases under the topographic melt pond scheme. Doing so removed the unrealistic higher-than-free run SIC RMSEs within the Siberian Chukchi Sea, and near-baseline performance in Coastal Canada. We ran a comparison of SIC RMSE reductions between both melt pond schemes to determine which scheme we should use by region in our results. The experiment revealed that we must use the level melt pond scheme for more accurate SIC representation in all regions except Coastal Canada and the Siberian-Chukchi Sea regions, where the topographic melt pond scheme statistically outperforms the level melt pond scheme for SIC representation across all assimilation configurations. This is given in Figure F below.

[Figure]

**Figure F:** Comparison of SIC RMSE for the topographic and level melt pond schemes. "T" indicates that the topographic scheme performed better, and "L" indicates that the level pond scheme performed better. The asterisk indicates significance; where "*" indicates $p < 0.1$ and "**" indicates $p < 0.05$.

Thus, for Coastal Canada and the Siberian-Chukchi Sea we opted to use the topographic melt pond scheme to reflect a more realistic melt pond fraction (and therefore albedo) for this location, keeping the more physical level melt pond scheme for the remaining regions. Such a more realistic melt pond fraction then led to improved assimilation performance across the board. Updated paper figures (Figs. B above and H below) are included below for emphasis of this change.

[Figure]

**Figure G.** Updated version of Figure 4 from the preprint, reflecting the addition of the Beaufort Sea region (as requested by Reviewer 3) and the switch to the topographic melt pond scheme for Coastal Canada and the Siberian–Chukchi Sea.

To further confirm that the topographic melt pond scheme better matched observed albedo observations in Coastal Canada and the Siberian Chukchi Sea, we compared our simulated experiments to satellite observations to illustrate an improved representation of ice-covered

albedo. Although the model contained a large negative bias in observed SIC, we compared the albedo *only* over ice-covered regions from satellite observations (using bilinear interpolation) with simulated albedos. Within the Siberian-Chukchi Seas, we see improved representation of albedo over ice-covered portions of the observed grid cell when using the topographic melt pond scheme (Figs. H and I). Thus, the authors feel confident in the chosen melt pond scheme for Coastal Canada and the Siberian-Chukchi Sea, despite being advised to not generally use this melt pond scheme in practice. We hypothesize that using full CICE (which includes a 2D dynamical representation of sea ice) would mitigate this degradation associated with the level melt-pond scheme by providing a more realistic dynamical evolution, thereby preventing near-unity coverage of level sea ice and the overrepresentation of melt-pond fraction.

[Figure]

**Figure H:** SIAL representation averaged seasonally (April 1 - October 15) over the free run period (2011-2015) for the model *Icepack* (blue line) and the satellite data (CLARA-A3 and NSIDC CDR) for Coastal Canada.

[Figure]

**Figure I:** SIAL representation averaged seasonally (April 1 - October 15) over the free run period (2011-2015) for the model *Icepack* (blue line) and the satellite data (CLARA-A3 and NSIDC CDR) for the Siberian-Chukchi Sea.

---

## Author Comment (AC3)

Review: Sea Ice Albedo Bounded Data Assimilation and Its Impact on Modeling: A Regional Approach

The paper presents a perfect model experiment comparing the prediction improvement caused by assimilating albedo in comparison to sea ice concentration and sea ice thickness in one dimensional Icepack simulations, for four regions in the Arctic. The study finds that depending on the error of the assimilated albedo and the region, that albedo assimilation outperforms SIC and SIT assimilation, except for one region. The work is relevant to the scope of The Cryosphere. Overall the paper is well structured and the experiments well designed and I would recommend minor revisions.

Thank you for sharing your overall favorable assessment.

**Main comments:**

1) The results emphasize the importance of the observation uncertainties. It is reasoned well why the SIC uncertainty varies with SIC, but should the same consideration not also be applied to SIT uncertainty and albedo uncertainty? I am unfamiliar with error dependencies in satellite albedo retrievals, but SIT errors are typically greater for thinner ice/thicker ice, depending on the product used. Since the errors are such a central part of the outcome of this study I would expect a higher focus on this.

Thank you for your insightful comment. Reviewer #2 raised a similar question regarding the uncertainty distributions and their influence on our results. Our choice of uncertainty configurations was guided by prior studies (e.g., Karellson et al., 2024 for SIAL; Wieringa et al., 2024 for SIC and SIT) and by discussions with Dr. Walt Meier at the National Snow and Ice Data Center (NSIDC). These reflect the distinct observational retrieval methods underlying each variable and their associated uncertainties. Specifically, SIAL retrievals are based on radiative transfer and depend on reflected shortwave radiative fluxes at the top of the atmosphere, whereas SIC retrievals rely on passive microwave measurements that exploit emissivity contrasts between open ocean and sea ice. SIT retrievals rely on a combination of both passive and active satellites, and its uncertainty is difficult to quantify. Because these retrieval techniques differ fundamentally, their uncertainty structures are also expected to differ.

To our knowledge, there are no widely accepted uncertainty constraints for any sea ice variable used in data assimilation—whether SIAL, SIC, or SIT. SIC is certainly the most developed and well-studied, given its widespread use in assimilation systems. For SIC, we used a negative parabolic uncertainty distribution with a maximum error centered at 12.5%. To test the sensitivity of this choice, we performed an additional experiment where the maximum SIC uncertainty was reduced to 5%, consistent with the "low" SIAL uncertainty configuration. The results, shown in Figure A, indicate that—except for the Barents Sea—this modification had little impact on the outcomes. We anticipated this limited sensitivity due to the bounded nature of

sea ice data assimilation: during transitional seasons (melt and freeze-up), SIC evolves rapidly toward its bounds, making the exact shape of its uncertainty distribution relatively inconsequential in comparison to avoiding filter divergence during transitional seasons.

[Figure]

**Figure A:** Demonstration showing that varying the maximum SIC uncertainty (12.5% vs. 5%) has minimal influence on assimilation outcomes (using the level melt pond scheme).

For SIAL, the seasonal evolution of albedo allows for more gradual transitions—from higher SIAL values to lower ones—driven by snow melt, persistent bare ice, and melt pond development. Thus, constraining SIAL uncertainty has a more noticeable effect on the results. The linear form of SIAL uncertainty reflects the dependence of retrieval error on viewing angle and the bidirectional reflectance distribution function (BRDF), which makes it harder to accurately observe bright surfaces such as snow and ice. Consequently, higher SIAL values tend to have greater absolute uncertainty, although the precise magnitude remains uncertain—this is an important topic for future research. Conversely, lower SIAL values (associated with open ocean or melt ponds) generally exhibit smaller retrieval errors. Although this was not stated explicitly in the manuscript, SIAL uncertainty from satellite retrievals is likely around 5–10% of SIAL. The largest uncertainty sources are cloud-masking errors and the effects of temporal averaging to mitigate BRDF impacts. Until these uncertainties are better constrained, comparing SIAL assimilation under different assumed uncertainty distributions (low, medium, high) remains necessary. Given the long satellite record of SIAL, our study emphasizes the value of systematically testing these uncertainty structures.

For SIT, we acknowledge that retrieval errors are typically larger for both very thin and very thick ice, though the exact magnitude depends strongly on the product used. Additionally, for Arctic SIT products, observations are typically only available in areas that are more than ~50% ice covered (depending on the product, e.g. CryoSat-2 vs. ICESat-2. This tends to screen out large regions of very thin ice in pan-Arctic sea ice thickness products. Quantifying SIT uncertainty is therefore challenging—both because of the limited observational coverage and

because each retrieval approach (e.g., radar altimetry, lidar) has different sensitivities. For this reason, we intentionally adopted a conservative approach by assigning SIT the lowest possible uncertainty distribution, comparable to the low-uncertainty configuration used for SIC. In this context we use the term 'conservative' to test our null hypothesis: that SIAL assimilation improves mean ice state prediction by assigning SIAL higher uncertainties and constraining SIC and SIT to low uncertainties. This enables a "best-case" comparison among SIC, SIT, and SIAL assimilation. Moreover, our use of a pan-Arctic daily assimilation window is highly idealized relative to the temporal resolution of the most accurate SIT products (e.g., ICESat-2), which provide only ~6 years of data and are difficult to validate given the scarcity of in-situ measurements. Because no universally accepted SIT uncertainty distribution currently exists, we chose this best-case configuration to avoid biasing our comparison in favor of SIAL assimilation.

Given the above information, we allude that the purpose of this study is to examine the additive effects of assimilating SIAL, and thus testing other uncertainty structures for SIC or SIT largely remains outside the scope of this study. It is of interest to the authors as potential future work.

2) One major issue in the paper is that the perfect model experiment is conducted in regions in which dynamics do play a non neglectable role in the ice grows. All experiment locations are located in regions which would probably yield different results, if conducted in a 3-D set up. This is not possible to fully address, but it would add to the significance of the study to include a site located in a region which is typically covered by land fast ice. If this would be the case it will be easier to relate future studies conducted in a 3-d set up to the findings in this study.

We thank the reviewer for this thoughtful suggestion. In response, we have added a fifth assimilation location in the Beaufort Sea to represent landfast ice. We selected this location because it has been the focus of extensive landfast ice research, owing to its proximity to Alaska's North Slope. The exact grid point used is shown in Figure B. To ensure that this location reflects landfast ice conditions, we turned off all dynamic processes by setting deformation forcing to zero (i.e., eliminating rafting and ridging parameterizations). This creates a point that is, in principle, analogous to a location in a three-dimensional model where dynamics are negligible.

[Figure]

**Figure B:** Locations of the discrete point-based regions used to spin up *Icepack* and to provide distinct ice conditions for evaluating the influence of different variable-assimilation configurations.

However, SIAL assimilation at this site resulted in degraded performance relative to SIC and/or SIT assimilation, and sometimes even the free run. We attribute this to the nature of landfast ice: it typically exhibits an abrupt transition from 100% SIC to 0% at a discrete boundary. Landfast ice is difficult for large-scale sea ice models to represent because of its small horizontal scale and largely thermodynamic behavior (Gu et al., 2022; Plante et al., 2024). Landfast ice is also difficult to resolve in observations at the satellite pixel scale similarly due to its small horizontal scale (Fraser et al., 2020).

In regions with active dynamics, SIAL assimilation tends to improve SIC and SIT RMSE because processes such as rafting and ridging introduce variability in surface properties that SIAL captures in correlation to SIC. In contrast, surface conditions in the Beaufort landfast region (based on this model configuration) are governed entirely by thermodynamics, which is much less correlated to SIC. To illustrate this, we include a correlation heatmap in Figure C during the melt season. The figure reveals that SIAL and SIC are not well correlated during the melt season within the Beaufort region, compared to the other seasonal ice locations. This suggests that SIAL is not an effective assimilation variable for SIC in this region—a result consistent with our findings in the Central Arctic, where SIC remains near 100% year-round and most assimilation configurations (outside of SIC-only assimilation itself) are similarly ineffective.

[Figure]

[Figure]

**Figure C:** Pearson correlation between SIAL and SIC in different *Icepack* modelled regions during the peak melt season (April 1 - October 15; SIC > 15% and SIAL > 0.01).

A key question is therefore why SIAL and SIC are only loosely correlated in our Beaufort experiments. Although we disabled lateral dynamic forcing to isolate purely thermodynamic evolution, the *Icepack* column still uses a slab-ocean formulation that permits basal heat exchange and melt, which is inconsistent with the mechanically constrained conditions required for grounded landfast ice. Therefore, we chose a location in the Beaufort Sea that is likely not bottomfast but instead part of the "stabilized fast ice" regime—floating ice that is only weakly anchored, if at all. The literature shows that the extent of Beaufort fast ice varies substantially year to year, and in some winters the region contains little or no fast ice at all. Thus, it is possible that our selected point lies outside the climatological landfast zone entirely, meaning the

slab-ocean setup may actually be more representative of free ice at this location than a more landfast behavior would be. These factors together likely contribute to the lower SIC–SIAL correlations in the model relative to observations. As shown in Figure D, observed correlations from 2011–2015 using NSIDC SIC and CLARA-A3 albedo are higher than those simulated, suggesting that either (1) Icepack's thermodynamic-only framework cannot reproduce Beaufort landfast behavior, (2) the selected site is not representative of persistent fast ice despite prior classifications (e.g., Jewell et al., 2025; Lange et al., 2025), or (3) SIAL assimilation provides limited benefit in regions where mechanical processes or interannual variability in fast-ice presence dominate seasonal evolution.

[Figure]

**Figure D:** Correlation between SIAL and SIC in satellite observations at the Beaufort Sea location (lat=75.54°, lon=174.45°; SIC > 15% and SIAL > 0.01) .

We therefore conclude that accurately representing Beaufort Sea landfast ice within our current *Icepack* configuration is challenging. The fine spatial scale of landfast ice, combined with JRA-55 atmospheric forcing and the absence of any dynamics, likely prevents *Icepack* from

reproducing realistic spatial and temporal behavior. For these reasons, we do not pursue further landfast ice experiments. Nonetheless, we include the Beaufort site and its results in the manuscript, noting the important caveat that this location may not be directly comparable to real landfast ice conditions, but that SIAL assimilation leads to degraded performance in the Beaufort Sea.

3) Currently the temporal set up is unclear. It reads as if the spin up is run for 11 years, the reference run for 5 and the assimilation for 7 months? A common description of which runs are run for how long would be nice for readability.

Thank you for this helpful comment. We agree that the temporal setup required clarification. In our experiments, the model was first spun up for 11 years to ensure equilibrium. The reference (free‑running) simulation was then integrated forward for 5 years, after which data assimilation was applied over a 7-month window (April–October 2011). We also performed assimilation experiments for subsequent years (2012, 2013, etc.) and found that although the absolute RMSE values varied somewhat by year, the relative performance of the different assimilation configurations (SIC, SIT, and SIAL) remained consistent. For clarity and brevity, we present results from the first year only in the manuscript. We will add this clarification around Line 90 to make the temporal setup more explicit.

4) The albedo is typically a parameter over which sea ice models are tuned. To ensure that the results are useful for future studies it would be desirable to attach the relevant namelist parameters. How would the authors expect the results to change if different values for, for example, snow grain size would be used?

Thank you for this helpful comment. We will include all albedo- and snow-related namelist parameters in the supplementary material as two tables for clarity and reproducibility, and have provided the relevant values below for your review.

Regarding the sensitivity of our results to albedo-related parameters: although parameters such as the snow grain effective radius are commonly tuned in sea ice models, we do not expect that varying them within physically reasonable ranges would meaningfully alter our conclusions. All albedo- and snow-related namelist parameters in our experiments use the **standard defaults recommended by the CICE consortium for CESM–CAM4 on the gx1 grid**, including the default snow grain radius parameter **R_snw = 1.5 μm**. This value lies squarely within the recommended range used across CESM and CICE configurations and is widely adopted in the broader community.

While some configurations adjust R_snw (e.g., toward smaller values or negative offsets) to fine-tune melt onset for particular atmosphere–ice couplings, such modifications primarily shift the **absolute magnitude** of simulated albedo rather than the underlying **covariance structure** among SIAL, SIC, and SIT. In our perfect-model assimilation framework, the relative

performance of assimilating SIC, SIT, or SIAL is determined primarily by these covariance relationships.

To evaluate this directly, we conducted sensitivity tests varying the snow grain radius across a wide range (-2–10 μm) in all five regions (Figs. E & F). These experiments led to only **minimal changes** in the simulated SIAL–SIC or SIAL–SIT correlations. Because these correlations are the dominant factor controlling assimilation outcomes, we expect that alternative but physically reasonable parameter choices would not change the relative behavior of the SIC-, SIT-, and SIAL-based assimilation experiments—only the overall magnitude of albedo, applied uniformly across experiments.

Accordingly, we are confident that our main results and conclusions are robust to typical parameter-tuning choices, including variations in snow grain radius and related albedo parameters.

[Figure]

**Figure E:** Impact of varying the snow grain radius on the SIC–SIAL correlation in *Icepack*. Changes in the correlation are minimal across the tested parameter range, except in the Beaufort Sea where correlation is relatively dependent on snow grain radius.

[Figure]

**Figure F:** Impact of varying the snow grain radius on the SIT–SIAL correlation in *Icepack*. As in Figure E, the resulting correlation differences are negligible, except in the Beaufort Sea where correlation is relatively dependent on snow grain radius.

**Icepack Model Configuration Parameter Tables**

| Parameter | *Icepack* Variable Name | Default Value | Units | Description |
|---|---|---|---|---|
| Snow grain radius | `r_snw` | 1.5 | unitless | Snow grain radius tuning parameter |
| Ice surface scattering layer | `hi_ssl` | 0.05 | m | Ice surface scattering layer thickness |
| Snow surface scattering layer | `hs_ssl` | 0.04 | m | Snow surface scattering layer thickness |
| Snow melt grain radius | `rsnw_mlt` | 1500.0 | kg/m²/s | Melting snow grain radius |
| Melt pond drainage timescale | `dt_mlt` | 1.0 | days | Drainage timescale for melt ponds |

**Table 1:** Albedo-Related Parameters that we specified in *Icepack* for our spinup, free run, and assimilation experiments.

| Parameter | *Icepack* Variable Name | Default Value | Units | Description |
|---|---|---|---|---|
| Snow thermal conductivity | `ksno` | 0.3 | W/m/K | Thermal conductivity of snow |
| Ice density | `rhoi` | 917.0 | kg/m³ | Density of sea ice |
| Snow density | `rhos` | 330.0 | kg/m³ | Density of snow |
| Ridging work parameter | `Cf` | 17 | unitless | Ratio of ridging work to potential energy change |
| Ice-ocean drag coefficient | `dragio` | 0.00536 | unitless | Drag coefficient at ice-ocean interface |

**Table 2:** Other sea ice tuning parameters relevant to our model setup in *Icepack*.

5) Overall the methods are well structured, including both the description of the experiment set up and the evaluation methods. It would further improve the structure of the paper to move the description of the category wise DA to the method section (4.3 to to example 2.5).

Thank you for this suggestion. We have moved the bulk of the category-wise DA description to the Methods section by adding a new subsection (Section 2.5), titled *"Category-wise Data Assimilation."* Specifically, we relocated lines 319–347 and made minor revisions to the wording to ensure the section fits naturally within its new position in the Methods.

**Minor comments:**

row 15: AA is only used once. No acronym needed.

Thank you for this comment. We have removed the acronym from our manuscript.

Line 27-31: missing reference

Thank you for your comment. We have added the appropriate references to cite that SIC and SIAL covary, especially during the melt season (Agarwal et al., 2011), and support of bare ice and open ocean albedo (Becker et al., 2023), as well as melt pond albedo (Grenfell and Perovich, 2004).

Some statements seem unnecessary for example line 41-46 and line 59-60. If the statements are not relevant to the study, why mention them. If the authors want to anyways include them, maybe these could be summarized to a motivation to study Arctic albedo processes.

Thank you for this helpful comment. We originally included these statements to emphasize the broader motivation for studying Arctic albedo processes. One of the main takeaways from this work is that uncertainty in sea ice albedo (SIAL) should be better quantified, and so the authors mention that understanding secondary processes—such as cloud and ocean feedbacks—is key to improving our representation of Arctic albedo evolution through reductions in observational uncertainty.

That said, we agree that the current wording is somewhat redundant and can be streamlined for clarity and focus. In the revised manuscript, these sentences have been condensed into a single motivating statement that better links the broader context to the objectives of this study. The revised text (centered around the original lines 41–46) now reads:

"Secondary processes such as changes in cloud cover and ocean temperature interact with sea ice melt to influence surface albedo and Arctic energy balance. Improving how these interconnected processes are represented in coupled sea ice simulations is essential for advancing our understanding of Arctic albedo evolution and reducing uncertainty in its role within the global climate system. The authors leave this as an open area of study for future research based on improved representation of Arctic sea ice."

Line 260: how the TRUTH was constructed should be in the methods

Thank you for this helpful comment. The construction of the TRUTH dataset is already described in the Methods section (lines 92–97). We have reviewed this portion to ensure the explanation is clear and sufficiently detailed, and for now are keeping the original text as shown below:

"For each assimilation experiment–defined as the set of simulations conducted for one of the five Arctic locations–a randomly selected ensemble member was designated as the reference TRUTH state, from which synthetic observations were derived for assimilation. To account for sensitivity to the choice of TRUTH, we repeated the assimilation experiments using ten different ensemble

members as TRUTH states (ensemble members 3, 5, 10, 12, 14, 16, 19, 21, 25, and 28). Synthetic observations of SIAL, SIC, and SIT were generated from each of these TRUTH realizations for assimilation into the remaining ensemble members."